# Phylogenomic profiles of whole-genome duplications in Poaceae and landscape of differential duplicate retention and losses among major Poaceae lineages

Taikui Zhang [1,2], Weichen Huang[1], Lin Zhang[2,4], De-Zhu Li [3], Ji Qi [2] ✉ & Hong Ma [1] ✉

Poaceae members shared a whole-genome duplication called rho. However, little is known about the evolutionary pattern of the rho-derived duplicates among Poaceae lineages and implications in adaptive evolution. Here we present phylogenomic/phylotranscriptomic analyses of 363 grasses covering all 12 subfamilies and report nine previously unknown whole-genome duplications. Furthermore, duplications from a single whole-genome duplication were mapped to multiple nodes on the species phylogeny; a whole-genome duplication was likely shared by woody bamboos with possible gene flow from herbaceous bamboos; and recent paralogues of a tetraploid *Oryza* are implicated in tolerance of seawater submergence. Moreover, rho duplicates showing differential retention among subfamilies include those with functions in environmental adaptations or morphogenesis, including *ACOT* for aquatic environments (Oryzoideae), *CK2β* for cold responses (Pooideae), *SPIRAL1* for rapid cell elongation (Bambusoideae), and *PAI1* for drought/cold responses (Panicoideae). This study presents a Poaceae whole-genome duplication profile with evidence for multiple evolutionary mechanisms that contribute to gene retention and losses.

Whole-genome duplication (WGD) events are identified as an evolutionary feature in many plants through genomic, phylogenomic, and phylotranscriptomic studies, especially in large angiosperm families (e.g., Asteraceae and Fabaceae) and larger clades (e.g., Myrtales and Asterids), and even across all angiosperm[1–6]. After WGDs, numerous retained gene duplicates provide raw genetic materials for evolutionary novelty, including diverse morphologies and adaptative changes that support great species richness[2,7]. In particular, analyses of synonymous substitution rate (Ks) values of gene duplicates placed 61

angiosperm WGDs on branches with increased diversification rates, suggesting the importance of WGDs in diversification[8]. Specifically, one of the duplicates from a WGD shared by Cucurbitaceae has been directly linked to the innovation of tendril formation[9], which is responsible for the climbing capacity of cucurbits. Furthermore, phylogenomic analyses of 25 angiosperm genomes support 14 WGDs at major phylogenetic nodes during geological periods with drastic environmental changes[10]. The retained duplicates are enriched for genes encoding transcriptional factor and components of regulatory

[1]Department of Biology, the Eberly College of Science, and the Huck Institutes of the Life Sciences, the Pennsylvania State University, University Park, State College, PA 16802, USA. [2]Ministry of Education Key Laboratory for Biodiversity Science and Ecological Engineering, School of Life Sciences, Fudan University, Shanghai 200438, China. [3]Germplasm Bank of Wild Species, Kunming Institute of Botany, Chinese Academy of Sciences, Kunming, Yunnan 650201, China. [4]Present address: Chongqing Key Laboratory of Plant Resource Conservation and Germplasm Innovation, School of Life Sciences, Southwest University, Chongqing 400715, China. ✉e-mail: qij@fudan.edu.cn; hxm16@psu.edu

networks related to stress response with possible roles in adaptation. Collectively, WGDs have directly resulted in variations of gene contents and are of great evolutionary importance in angiosperms. However, differential retention and loss of duplicates in separate lineages that share the same WGD(s) are much less explored.

Poaceae are the fifth-largest family (~12,000 species in 12 subfamilies) and the core Poaceae comprise two clades named PACMAD and BOP[11–13]. The PACMAD clade includes subfamilies Panicoideae, Chloridoideae, Danthonioideae, Arundinoideae, Micrairoideae, and Aristidoideae. The BOP clade consists of subfamilies Pooideae, Oryzoideae, and Bambusoideae. Poaceae include numerous economically important species in Panicoideae (maize and sorghum; the second largest subfamily), Chloridoideae (teff), Pooideae (wheat and barley; the largest), Oryzoideae (rice), and Bambusoideae (bamboos, the third largest) (e.g., refs. [11–13]). Grasses have diverse morphologies; for instance, most bamboos are woody, in contrast to herbaceous for most grasses[12]. Also rice and other Oryzoideae members grow in fresh and salt-water aquatic environments rather than in dry ecosystems for most other grasses[14]. In addition, wheat and most other Pooideae members are adapted to cold and cool environments at high latitudes and altitudes, but grasses with C4 photosynthesis in Panicoideae and Cloridoideae are distributed in areas with hot and dry environments[12]. Specifically, ~40% of the earth's land surface are grasslands and bamboo forests, and ~60% of C4 plant species are grasses[12,13]. The great Poaceae diversity provides an excellent system to investigate the possible evolutionary impact of WGDs and differential duplicate retention and loss among different subfamilies.

WGDs can be strongly supported by chromosomal collinearity (synteny); however, ancient WGDs might lack clear syntenic signals due to gene loss and possible genome rearrangements, especially following more recent WGD(s)[4]. Complementary to syntenic analyses, evidence for WGDs can also be obtained from multiple gene duplications (GDs) or GD clusters mapped to specific nodes in a species phylogeny by comparing gene phylogenies with species-tree and by molecular dating estimates of paralogous gene pairs relative to those of orthologs from investigated taxa; such evidence can be retrieved from one genome for different WGDs[1,4]. Thus, clusters of syntenic paralogues from a large fraction of the genome are considered as strong evidence for WGDs with phylogenetic placements or age estimates.

In Poaceae, synteny studies and molecular dating indicated that Poaceae members share three ancient polyploidizations, including the tau WGD shared by most monocots, the sigma triplication shared by the order Poales, and the Poaceae-specific rho WGD[15–17]. Also, rho has been supported by chromosomal collinearity in members of early-divergent grass subfamilies (Anomochlooideae and Pharoideae) and the core Poaceae[18,19]. Several subsequent WGDs in five subfamilies (Bambusoideae, Pooideae, Panicoideae, Chloridoideae, and Oryzoideae) are supported by genomic or phylogenomic analyses (e.g., refs. [20–26]). However, it is not known whether five other subfamilies (Aristidoideae, Micrairoideae, Arundinoideae, Danthonioideae, and Puelioideae) or subclades of large subfamilies without sequenced genomes have lineage-specific WGDs. Moreover, woody bamboos are polyploids and were proposed to have resulted from hybridizations among four hypothesized diploid ancestors (subgenomes) with one of the subgenomes shared by extant woody bamboos[23]; but the early bamboo genome evolutionary history is still unclear. Additionally, rice and other *Oryza* species collectively have 11 reported genome types (six diploids and five allotetraploids)[27]. Domestic and wild *Oryza* species have adapted to different aquatic environments[14]. Specifically, available sequenced *Oryza* genomes for several diploids and an allotetraploid[28–33] provide an opportunity to investigate the origins of the allotetraploid and possible contributions of subgenomes to the adaptations to high salt and submerged aquatic environments.

Following WGDs, chromosomal rearrangements and gene loss (fractionation) can result in a diverse landscape of gene copy number variations across gene families and species[34,35], in part due to lineage-specific gene retention and losses. This idea is supported by the detection of GD clusters at successive nodes on species trees in phylogenomic studies[1,9,36]. For instance, two GD clusters were detected at successive nodes on the Ericales phylogeny and shared by most families, and chromosomal collinearity supports a WGD event corresponding to the deeper of the two nodes[1]. Furthermore, differential loss of duplicates from WGD among subclades can lead to reproductive isolation and contribute to speciation[37]. Comparison of gene contents of three yeast species that shared a WGD revealed >200 genes that experienced differential losses resulting in a single copy in each species, with >180 other genes showing different patterns of retention/loss, leading to 4–7% of single-copy genes between any two species being paralogs rather than orthologs[38]. Moreover, retained duplicates from WGDs can experience different forms of functional differentiation, including neofunctionalization and subfunctionalization, often under differential selection[35,38]. Systematic and comprehensive integration of genome syntenic information and phylogenomic results of genes from multiple species can provide insights into genome evolution; however, such analysis has been limited, in part due to the scarcity of plant groups with (1) multiple sequenced genomes, (2) many large gene sequence datasets (such as transcriptomic datasets), and (3) large-scale species phylogenies.

Previously rho was linked to an up-shift of diversification rate in early Poaceae[8], suggesting a contribution of rho to grass species diversification. In addition, 411 rho-derived GDs were mapped to the most recent common ancestor (MRCA) of Anomochlooideae and the core Poaceae, and 123 rho-derived GDs were mapped to the core Poaceae[36]. Differences in lineage-specific retention from rho were observed in several gene families (e.g., MADS-box) from comparisons of as many as seven genomes in six or fewer grass subfamilies[18,19]. Specifically, phylogenetic analyses of *indeterminate spikelet1* homologues placed a GD at the origin of Poaceae and two Anomochlooideae copies as successive sisters to the gene clade of core Poaceae homologues, suggesting that this gene might have affected floral phenotypes differently between Anomochlooideae and the core Poaceae[19]. In addition, some rho-derived duplicates were shown to have different functions in grass development and stress response. For example, the rice rho-derived paralogs *MADS50* and *MADS51* act upstream of the *Early heading date1* gene to regulate flowering transition, but are differentially regulated by histone methylation[39,40]. Furthermore, one copy (*LOC_Os01g66100*) of the rice *SD1* genes (but not the other) is involved in gibberellin biosynthesis and promotes internode elongation in plants grown in deep-water, supporting neofunctionalization responsible for adaptation to periodic flooding[41]. However, genome-wide analyses of the number and potential functional differentiation of rho-derived gene duplicates have not been conducted to detect retention and loss patterns in different subfamilies, largely because genome sequences for several Poaceae subfamilies were not available until very recently.

We used nuclear genes from genomic/transcriptomic datasets of >360 grasses to reconstruct a Poaceae phylogeny with well-resolved phylogenetic relationships among subfamilies and tribes[13] (Supplementary Fig. 1). This Poaceae nuclear phylogeny and the >360 genomic/transcriptomic datasets provide an excellent opportunity to investigate WGDs in multiple subfamilies, to identify potential WGDs across Poaceae, to investigate relationships between GD clusters at successive nodes and WGDs, to detect evidence for potential hybridizations, and to explore lineage-specific retention of rho-derived gene pairs and those from other WGDs in grasses. Here our phylogenomic analyses generate a landscape of WGDs across Poaceae and other Poales lineages, providing phylogenetic placements of WGDs previously supported by analyses of a few species and reporting

previously unknown WGDs. We further present an investigation of GDs supported by syntenic regions from rho or other WGDs within specific subfamilies (Bambusoideae and Oryzoideae) and report their association with two or more phylogenetically placed GD clusters, with insights on the evolution of gene duplicates. Finally, we examined the genome evolution during the Poaceae history, focusing on the patterns of differential retention and losses of rho-derived gene duplicates among representatives of major subfamilies, with implications for lineage-dependent functional diversification and adaptive divergences of grasses. Our results support lineage-specific WGDs, differential sequence evolution by gene conversion, and specific duplicate retention and/or loss as likely mechanisms for the differential impact of WGDs on Poaceae gene function and species diversification.

## Results

### WGDs identified in Poaceae and other Poales lineages

Among the published 349 datasets (342 transcriptomes and seven genome-skimming datasets) generated for our previous grass phylogenomic/phylotranscriptomic studies[13,20], we selected 319 datasets (315 transcriptomes and four genome-skimming datasets) for our analyses here (Supplementary Data 1). WGD analyses included the use of gene- and species-tree reconciliation using Tree2GD and Ks analyses (see species-tree in Supplementary Figs. 1, 2, and more details in methods), taking advantage of the recently established Poaceae/Poales phylogenies[13]. Additional datasets for 53 Poaceae, 17 for other Poales, and 10 for other orders were retrieved from public databases (see taxon and transcript assembly information in Supplementary Data 1, 2, respectively). We identified GD clusters (Supplementary Figs. 3–16; see methods) that support 22 proposed WGD events (#1–22; Fig. 1). For the WGDs supported by gene duplicates from sequenced Poaceae genomes, we further estimated the number of GDs with detected duplicates in syntenic blocks. The 22 proposed WGDs were assigned into three types, including nine previously unknown WGDs (#6, 10, 13, 15, 16, and 19–22), four reported WGDs with different phylogenetic positions here (#4, 5, 7, and 8), and nine WGDs (#1–3, 9, 11, 12, 14, 17, and 18) consistent with previous reports[2,15–17,26,42,43].

Previously unknown WGDs and others are placed onto a species-tree and described here. The nine previously unknown WGDs in Poaceae here include six genus-specific WGDs (#6, 10, 13, 16, 21, and 22) in five subfamilies: Puelioideae I (#6 *Puelia*, Supplementary Fig. 4), Pooideae (#10 *Avena*, Supplementary Fig. 8), Panicoideae (#13 *Ischaemum*, Supplementary Fig. 13), Danthonioideae (#16 *Danthonia*, Supplementary Fig. 4), and Chloridoideae (#21 *Eleusine*, Supplementary Fig. 14; #22 *Perotis*, Supplementary Fig. 16). The WGD (#10) shared by *A. sativa* (2n = 6x = 42) and *A. barbata* (2n = 4x = 28) is associated with the polyploid evolution of *Avena* species, which comprise diploids, tetraploids, and hexaploids[44]. In addition, three WGDs (#15, 19, and 20) are each shared by two genera, one in Panicoideae (#15 shared by *Andropogon* and *Schizachyrium*, Supplementary Fig. 13) and two in Chloridoideae: at the MRCA of *Tridens brasiliensis* and *Pappophorum vaginatum* (#20; Supplementary Fig. 15) and the MRCA of *Tridentopsis mutica* and *Gouinia latifolia* (#19; Supplementary Fig. 16). Furthermore, four previously proposed WGDs[2,22,25] are supported by the analyses here, which placed them at different phylogenetic positions using data from additional related species (#4, 5, 7, and 8; Fig. 1). In the Poales family Typhaceae, a *Typha* WGD (#4; Supplementary Fig. 3) is shared by *T. orientalis*, *T. latifolia* and *T. angustifolia*, consistent with a previously identified WGD shared by *T. latifolia* and *T. angustifolia* (referred to as TYPHα[2]). Our results also placed a WGD at the MRCA of Cyperaceae and Juncaceae (#5; Supplementary Fig. 3), which seems the same as the JUINα in *Juncus*, the LEGIα in *Lepidosperma gibsonii* and the CYPAα in *Cyperus*[2]. In Oryzeae, the WGD detected in the *Zizania latifolia* genome[25] was placed at the MRCA of *Z. latifolia* and *Rhynchoryza subulata* (#7; Supplementary Fig. 6). In Bambusoideae, the WGD supported in *Phyllostachys edulis* genome[22] is

placed at the MRCA of woody bamboos (#8; Fig. 1; see below for more results on this WGD). Finally, nine WGDs detected here are consistent with those supported by previous studies[2,20,26,42,43,45,46], including tau shared by the core monocots after the divergence of Alismatales (#1; Supplementary Fig. 3), sigma in Poales (#2; Supplementary Fig. 3), rho (#3; Supplementary Figs. 3, 4), and six other WGDs (#9, 11, 12, 14, 17, and 18; Supplementary Figs. 8, 9, 11, 13, 16, 14).

The above WGDs were also supported by evidence from Ks peaks. The Ks among paralogues has been widely used as a correlate of relative time for the divergence of paralogues; when Ks values form a peak in a distribution, the corresponding GDs are considered to be in a cluster near a specific time and used as support for WGDs[1,2]. For example, the OneKP study has used the detection of Ks peaks among paralogues from separate analyses of sequences of 99 single species as support for 99 WGDs in plants[2]. Thus Ks was analyzed for paralogues identified here (see methods), and Ks peaks shared by multiple species were observed, providing additional support for WGDs from the Tree2GD analyses (Supplementary Figs. 17–23 and Supplementary Data 3). In particular, the Ks peak of paralogues from a proposed WGD in a focal species is expected to have a higher value than that of orthologues between the focal species and its closely related species that also shares the WGD, and lower than that of orthologues between the focal species and an outgroup species, which diverged before the WGD event. For example, the previously unknown WGD for *Ischaemum* (WGD#13; Supplementary Fig. 21 and Supplementary Data 3) is supported by the Ks peak value of 0.1144 for paralogues mapped at the MRCA of two *Ischaemum* species; this Ks value is higher than the Ks peak value (0.0599) of orthologues between the two *Ischaemum* species, but lower than the Ks peak value (0.1184) between *I. aristatum* and the outgroup *Eulaliopsis binata*. The *Ischaemum* WGD and other previously unknown WGDs here provide a resource for analyses of genome evolution in grasses and can be strengthened by future analyses using greater taxon sampling and genome sequencing.

Due to differential evolutionary rates, variation in Ks peak values of duplicates in different taxon lineages from the same WGD has been observed in several dating analyses of WGDs[1,2]; for instance, different Ks values of the rho-derived duplicates were reported in different grasses[2,18,36]. To further estimate the difference for rho, we surveyed the evolutionary rate (estimated by branch length) between species and the Ks value of retained paralogs from each species. Our results indicate that Ks values are positively correlated with the total branch length from the Poaceae MRCA to tips (Coefficient: 0.89, *p* value = 1.21e-08) (Supplementary Fig. 24). Hence, WGD dating can be affected by the different evolutionary rates of species, including the accelerated (e.g., Panicoideae species) or reduced (e.g., Bambusoideae species) mutation rates. Thus a higher Ks peak value in a rapidly evolving lineage after a WGD compared to the Ks peak value of an outgroup that diverged before the WGD could incorrectly place a WGD at an earlier node. When paralogues with differential evolutionary rates are included in phylogenomic analyses, GDs from small-scale duplications (SSDs, such as tandem duplications) could also be detected as "clusters" and treated as evidence for WGDs[47]. The MAPS method was developed to detect the effect of variations in branch lengths on GD signals[48] and was used here to analyze gene duplicates of sequenced grass genomes with WGD signals. The results indicated that the rho and tau events were supported (*p* value <0.001; Supplementary Fig. 25) by MAPS analyses. Nevertheless, candidate WGDs identified by using Ks analysis should be further tested and strengthened using phylogenomic and syntenic approaches.

Our phylogenomic analyses also detected eight other GD clusters (#23-30, Supplementary Figs. 6, 7, 12, 13), in addition to the above WGD events (#1–22; Fig.1). These GD clusters are each shared by species with sequenced genomes but lack sufficient synteny support; they might correspond to ancient SSD events in the MRCA of the affected clades. For example, a GD cluster (#23, Supplementary Fig. 12) was detected at

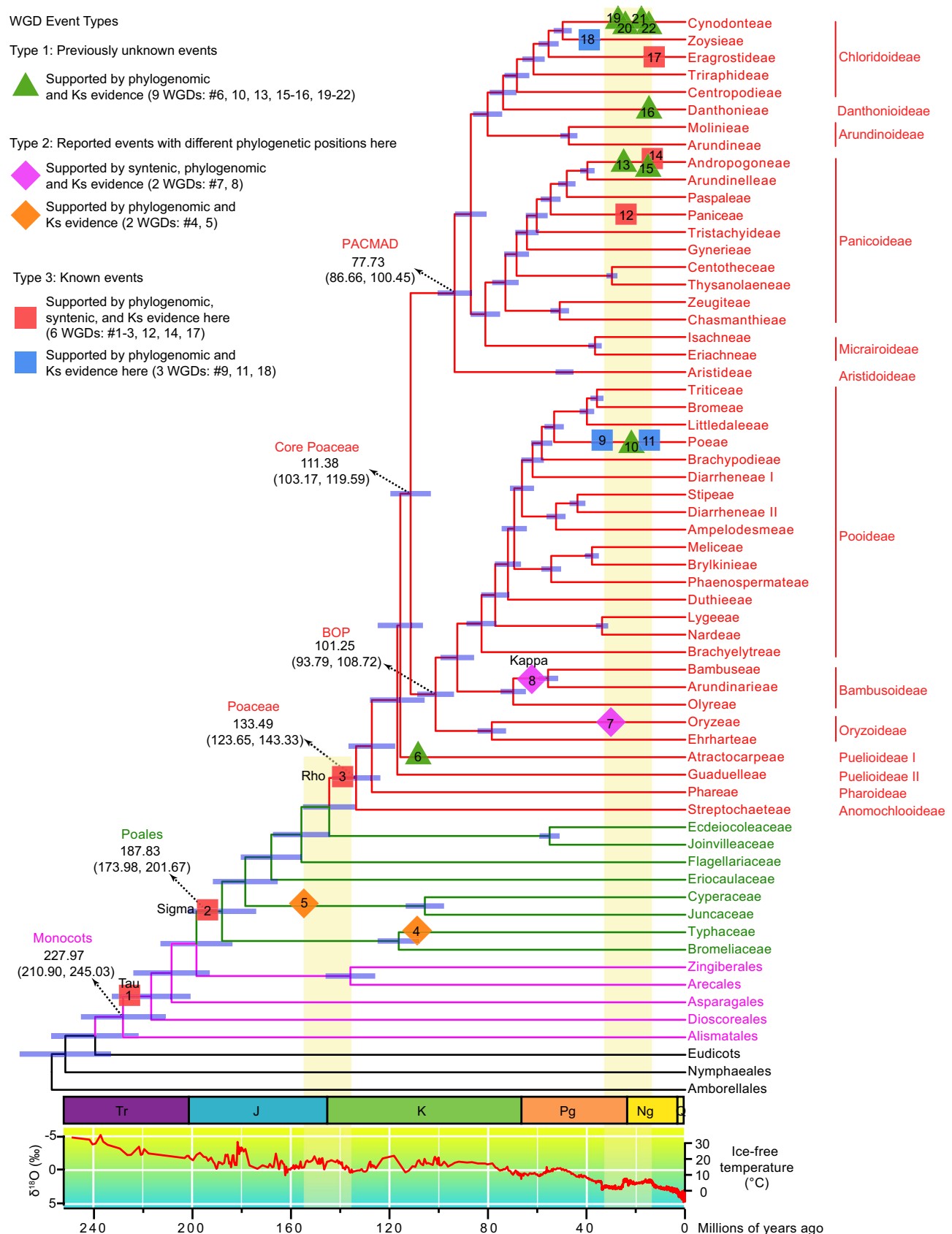

the MRCA of two Panicoideae supertribes, Andropogonodae (with *Sorghum bicolor* and *Zea mays*) and Panicodae (with *Cenchrus, Panicum,* and *Setaria*). However, examinations of genomes of members of Andropogonodae and Panicodae did not detect syntenic support for

GDs in this cluster. Instead, many of the paralogues supporting GD cluster #23 matched tandem repeats (Supplementary Fig. 12), including 189 of 473 GDs in *Sorghum bicolor* and 59 of 279 GDs in *Zea mays*; similarly, 103 of 236 GDs in *Cenchrus americanus* and 185 of 433 GDs in

**Fig. 1 | Identification of WGD events in Poaceae and other Poales families.** A simplified Poaceae/Poales phylogeny (from the detailed tree in Supplementary Fig. 1) is shown with branch length representing the median estimated time (see Supplementary Fig. 2) from divergence to the present. Red lines in the tree represent Poaceae tribes, with the tribe names to the right of the terminal branches, and subfamily names further right. Other Poales families are shown as green branches and names. Other monocot orders are shown in pink. Horizontal blue bars at each node indicate the 95% credible interval of divergence time in millions of years. For major groups (indicated by dotted arrows), the numbers below the names and numbers within round brackets indicate the median value and the 95% confidence interval of divergence time millions of years ago, respectively. Stratigraphic periods of Triassic (Tr), Jurassic (J), Cretaceous (K), Paleogene (Pg), Neogene (Ng), and Quaternary (Q) are illustrated by colored boxes below the tree. WGD events are placed on branches with approximate divergence times and marked by numbered triangles, squares, or rhombuses in five different colors (See Supplementary Figs. 3–16 for detailed positions of WGDs; See Supplementary Figs. 17–23 and Supplementary Data 3 for detailed dating of WGDs). Green triangles, previously unknown WGDs. Pink rhombuses, WGDs that have different phylogenetic positions here with support from syntenic, phylogenomic, and Ks evidence. Orange rhombuses, WGDs that have different phylogenetic positions here with support from phylogenomic and Ks evidence. Red squares, WGDs with support from syntenic, phylogenomic and Ks evidence here and consistent with previous reports. Blue squares, WGDs with support from phylogenomic and Ks evidence here and consistent with previous reports. The red curve in the graph below the stratigraphic boxes illustrates the changes in oxygen isotope records of δ18O (‰) (the left Y-axis), reflecting the temperature changes as indicated by the right Y-axis. Left and right vertical yellow bars, respectively, indicates the J-K and Pg-Ng boundary with a 10 million years flanking period on either side. Source data are provided in a Source Data file.

*Setaria italica* correspond to tandem duplicates. It is possible that gene duplicates from ancient SSD events, including tandem duplications, have promoted the extensive divergence in Andropogonodae and Panicodae, which together account for ~96% of the Panicoideae species diversity[11]. Further support for this idea is provided by the detection of other GD clusters with ancient tandem duplications within this large clade: at Panicodae (#24, Supplementary Fig. 12), at the MRCA of most Andropogoneae subtribes (#25, Supplementary Fig. 13), and at Andropogonodae (#26, Supplementary Fig. 13). Other GD clusters were also found at the MRCA of Poeae and Meliceae tribes (#27, Supplementary Fig. 7), at Pooideae (#28, Supplementary Fig. 7), at Oryzinae, and at Oryzeae (#29, #30, Supplementary Fig. 6).

## Syntenic analyses of successive GD bursts after rho and evidence for gene conversion

Our above phylogenomic analyses detected 1,633 GDs mapped at Poaceae, corresponding to the rho event (#3; Fig. 1). Along the backbone of Poaceae (Fig. 2a), three other GD bursts were successively observed at the MRCA of Pharoideae and other grasses (151 GDs, C3), the MRCA of Puelioideae II and others (409 GDs, C2), and the MRCA of the core Poaceae (936 GDs, C1). Similar duplication patterns are also observed in the MAPS results (Supplementary Fig. 25). These GDs might be related to differential retention and loss from earlier duplication event(s), such as rho, or due to SSDs (as described above); to test these possibilities, we examined the paralogues that correspond to these GDs in representative sequenced genomes for their presence in syntenic blocks (Fig. 2b, c). For the detected GDs at the MRCA of two lineages (A and B), we classified the gene tree topologies into three types: (AB)(AB), (AB)A, and (AB)B. Here the (AB)(AB) type means that both paralogues were retained in both lineages A and B, and multiple GDs of this type are considered evidence of WGD. On the other hand, the (AB)A and (AB)B retention types would represent the loss of one paralogue in the B or A lineages, respectively.

For the 1633 GDs mapped at the MRCA of Poaceae, 744, 546, and 343 GDs belong to, respectively, the (AB)(AB), (AB)A, and (AB)B types (Supplementary Fig. 26a), where the A lineage is the clade from Pharoideae to the core Poaceae, with the B lineage being Anomochlooideae. Furthermore, we examined the GDs-derived paralogues for their positions in syntenic blocks in sequenced grass genomes and found syntenic paralogs in at least one species for 605 of the 744 (AB)(AB)-type GDs, showing evidence for the rho event (Supplementary Fig. 26a). Furthermore, 487 of the 546 (AB)A-type GDs and 72 of the 343 (AB)B-type GDs matched paralogues in syntenic blocks (Supplementary Fig. 26a); it is worth noting that 454 of the 487 syntenic GDs and 33 of the 72 syntenic GDs are in the collinear genomic blocks (Supplementary Fig. 26b) that also have some GDs of the (AB)(AB) type (portion of the 744 GDs). These results indicated the many GDs of the (AB)(AB), (AB)A, and (AB)B types are from the same event (rho) (Supplementary Fig. 26b). For example, phylogenetic trees of the two

rice gene pairs anchored in a syntenic block from rho indicate that one pair (*LOC_Os12g42570*, *LOC_Os03g44670*) corresponds to a GD at the MRCA of Poaceae of the (AB)(AB) type (Supplementary Fig. 26c, d) and the other pair (*LOC_Os12g42260*, *LOC_Os03g44310*) matches a GD at the MRCA of Poaceae with the (AB)A retention type (Supplementary Fig. 26e), suggesting the pair of rice genes retained in (AB)A type likely correspond to a gene loss event in the B lineage (Anomochlooideae) after rho.

Furthermore, to assess the three GD bursts that were placed at successive nodes shared by multiple Poaceae subfamilies (C1–C3, Fig. 2a, and Supplementary Figs. 3, 27–29), we also classified GD retention types in gene trees. Our results revealed that, for the 936 GDs in C1, 699, 106, and 131 GDs, respectively, were the (AB)(AB), (AB)A, and (AB)B types (Supplementary Fig. 27a). To further investigate whether the three types of GDs in the cluster C1 were from a single WGD event, we analyzed the paralogues corresponding to these GDs in sequenced grass genomes and found that 280 of the 936 GDs include syntenic genes in one or more species, including 231 (AB)(AB)-type GDs (Supplementary Fig. 27a). Among the 231 GDs, 91 are anchored in the syntenic blocks that also contain genes of the (AB)A or (AB)B types. In addition, 190 GDs (C1) at the MRCA of core Poaceae were mapped to syntenic blocks that also contain genes corresponding to the GDs mapped at the MRCA of Poaceae (rho) (Supplementary Fig. 27a). For example, the rice paralogues of *LOC_Os03g43880* and *LOC_Os12g41720* (see synteny in Supplementary Fig. 26c and gene tree in Supplementary Fig. 27b) are mapped at the core Poaceae; the same syntenic block also has gene pairs mapped to the MRCA of Poaceae (Supplementary Fig. 26c). The observation that the same syntenic blocks contain paralogues of both GDs mapped to the MRCA of Poaceae and those of core Poaceae support the idea that these GDs corresponded to the same WGD, the rho, event. Among the 190 GDs mapped at the core Poaceae with syntenic gene pairs and proposed to have been derived from rho, 147 are supported by the outgroup from Anomochlooideae and/or Pharoideae (Supplementary Fig. 30a).

In addition, we also examined paralogues of the GD bursts (C2-C3) mapped at two other inter-subfamilial positions before the MRCA of core Poaceae for possible syntenic evidence that some of them were likely from rho (Supplementary Figs. 28, 29). For the C3 cluster with 151 GDs, 123 (81.46%) of GDs were of the (AB)(AB) type and 28 (18.54%) were of the (AB)A type (Supplementary Fig. 28a). Synteny analyses showed that 91 GDs with the (AB)(AB) type include syntenic genes and the syntenic gene pairs for 82 of these 91 GDs are located in the syntenic blocks that also contain genes mapped at the MRCA of Poaceae (rho) (Supplementary Fig. 28a). The rho-derived anchored pairs in a synteny block with different phylogenetic positions provide an association of the successive GD clusters with the rho event. Some of the gene trees lacked the rho-derived paralogues from Anomochlooideae, suggesting that both gene duplicates were lost in this subfamily, resulting in the placement of the GDs in cluster C3. An example for

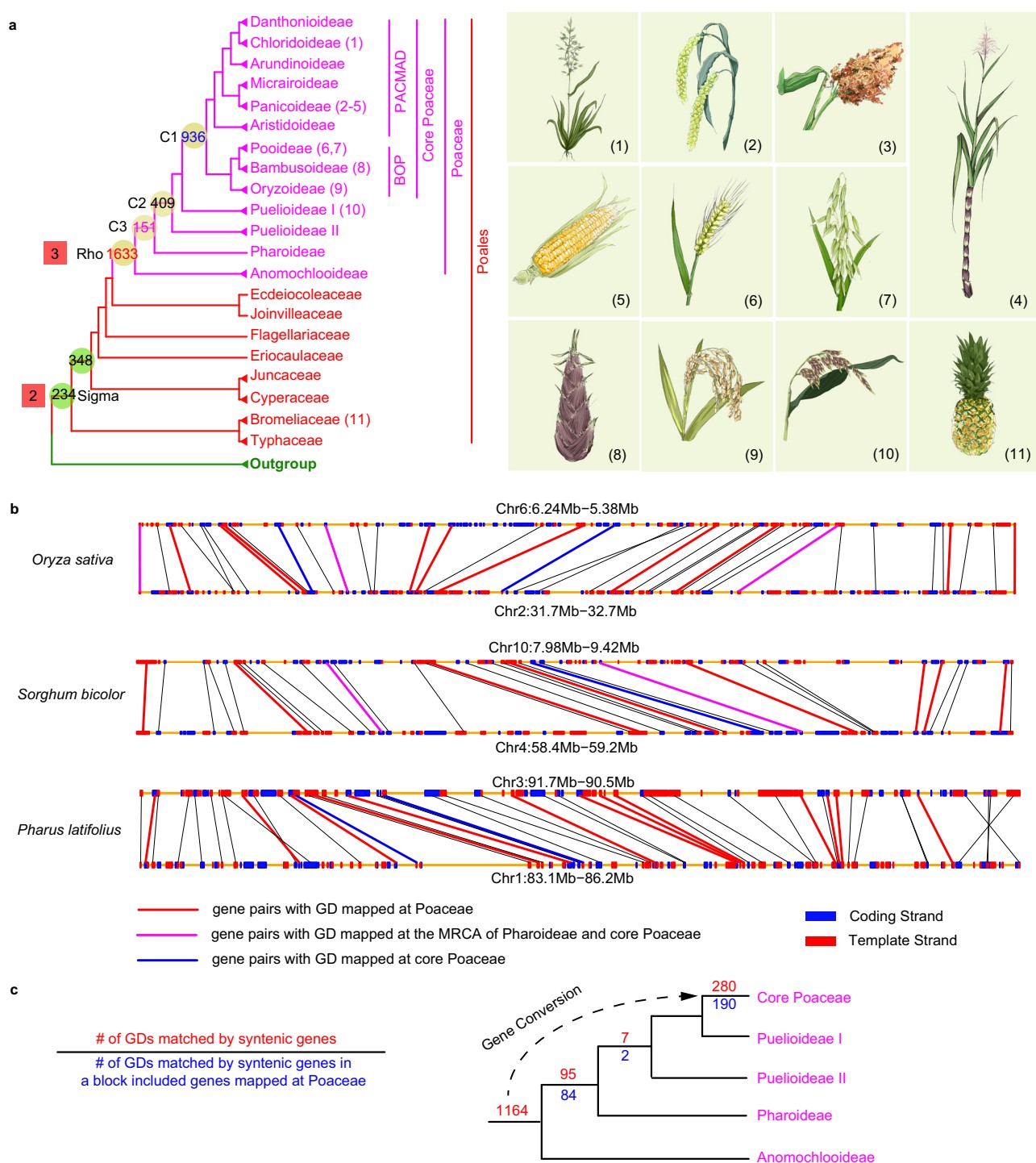

**Fig. 2 | Successive GD clusters in Poaceae and their relationship to rho. a** (left) Number of GDs shared by grasses (pink branches) and other Poales (red branches). For the rho event and the successive GD clusters (C1–C3) additional information is shown in Supplementary Figs. 26–29, including the numbers of GDs anchored in genome collinear blocks) detected in individual species with different retention types; (right) drawings illustrating ten representatives grasses (1-teff, *Eragrostis tef*; 2-foxtail millet, *Setaria italica*; 3-sorghum, *Sorghum bicolor*; 4-sugarcane, *Saccharum spontaneum*; 5-maize, *Zea mays*; 6-wheat, *Triticum aestivum*; 7-oat, *Avena sativa*; 8-bamboo, *Phyllostachys edulis*; 9-rice, *Oryza sativa*; and 10-*Puelia ciliata*) and pineapple (*Ananas comosus*; 11; representing Bromeliaceae). **b** Illustrations of chromosomal collinearity (synteny) for gene pairs in three grass representatives (*O. sativa*, *S. bicolor*, and *P. latifolius*). Lines between different chromosome segments represent syntenic relationships, some of which are linked by thick and colored lines indicating different phylogenetic positions of the duplication shared by the genes, as indicated below the blocks. Black lines represent syntenic genes not mapped to the three positions. Blue and red rectangles represent protein-coding genes with coding or template strands, respectively. **c** Number of GDs matched by syntenic genes in Poaceae. The red number above the branch represents the number of GDs matched by syntenic genes; the blue number below the branch represents the number of ones in synteny blocks that have some other gene pairs mapped at Poaceae; specifically, 190 of 280 GDs mapped at the core Poaceae are in syntenic blocks with gene pairs from rho and might be retained at the core Poaceae through gene conversions. A proposed model of gene conversion can be found in Supplementary Fig. 30. Source data are provided in a Source Data file.

such differential retention (in Pharoideae and the core Poaceae) and loss (in Anomochlooideae) is the rice gene pair (*LOC_Os12g42310* and *LOC_Os03g44500*; Supplementary Figs. 26c, 28b), which corresponds to a GD mapped at the MRCA of Pharoideae and the core Poaceae and is placed in a syntenic block with some GDs mapped at Poaceae. Similarly, the loss of both duplicates in all three basal grass subfamilies can result in the gene topology of ((core Poaceae, core Poaceae), non-Poales) (Supplementary Fig. 30a for two GDs).

On the other hand, gene topologies of 35 GDs with putative rho-derived syntenic genes mapped at the MRCA of Pharoideae and the core Poaceae are supported by the outgroup from Anomochlooideae (with possible loss of one Anomochlooideae gene) (Supplementary Fig. 30b). The observation that anchor genes in the syntenic blocks linked to rho were mapped to both the MRCA of Poaceae and subsequent backbone nodes of the grass phylogeny suggested that, following the rho event, the sequences of the paralogues of an ingroup (such as the core Poaceae) became more similar to each other, relative to sequences of earlier divergent lineage(s) (such as Anomochlooideae). This could be due to gene conversion (also referred to as nonreciprocal exchange), which copies the sequence of one homologue to replace the sequence of another during meiosis (and, to a lesser extent, during the mitotic cell cycle) and can lead to equalization of different gene copies[49]. Indeed, analyses of diploid and allopolyploid cottons (*Gossypium*) supported gene conversion between homologous sequences from past WGD[50]. In addition, a comparison of *Brassica rapa* and *B. oleracea* sequences supported gene conversion of 368 and 343 syntenic genes, respectively[51]. In grasses, previous Ks analyses of rho-derived paralogues and their corresponding orthologues from five grasses [two rice subspecies, another member of Oryzoideae, *Brachypodium* (Pooideae) and *S. bicolor* (Panicoideae)] found evidence for likely gene conversion of 58 paralogous pairs after the divergence of rice and *S. bicolor*[52]. This study lacked the three early-divergent grass subfamilies, and thus did not describe gene conversion before the MRCA of core Poaceae. Here the rho-derived syntenic gene pairs mapped at the core Poaceae with early-divergent Poaceae as their closest outgroup(s) (Supplementary Fig. 30a) and at the MRCA of Pharoideae and the core Poaceae with Anomochlooideae as their closest outgroup (Supplementary Fig. 30b) are placed in syntenic blocks where some gene pairs matched GDs at Poaceae, providing insights into gene conversion as part of post-WGD gene evolution (Fig. 2c; see Supplementary Fig. 30c for a model of gene conversion).

## A proposed paleo-polyploidization of woody bamboo ancestor

The bamboo subfamily Bambusoideae contain diploid herbaceous bamboos (HB = the Olyreae tribe; $2n = 2x = 20–24$) and polyploid woody bamboos (WB), with the Arundinarieae (tetraploid temperate bamboos; $2n = 4x = 46–48$) and Bambuseae tribes, which include tetraploid neotropical ($2n = 4x = 40–48$) and hexaploid paleotropical ($2n = 6x = 70–72$) bamboos[11,23]. The Arundinarieae tetraploidy was supported by extensive collinearity in the sequenced *Phyllostachys edulis* genome[22]. In addition, a phylogenetic study[53] of three nuclear genes from 36 bamboo species supported a proposed 5-subgenome model (A, B, C, D, and E) of WB subgenome types: AABB for Arundinarieae, CCDD for the tetraploid Bambuseae and CCDDEE for the hexaploid Bambuseae. Among the five subgenomes, B and C are relatively close. More recently, a genome-scale comparison of two HBs, one Arundinarieae species, one tetraploid Bambuseae, and two hexaploid Bambuseae provided support for a revised model with redefined A, B, C, and D subgenomes[23]. The A subgenome is specific to hexaploid Bambuseae, B is shared by the tetraploid and hexaploid Bambuseae, C is shared by all WBs, whereas D is specific to Arundinarieae[23]. Thus the hexaploid Bambuseae, the tetraploid Bambuseae, and Arundinarieae have, respectively, AABBCC, BBCC, and CCDD genomes. Our above phylogenomic analyses covered five Olyreae genera, 14 genera of Arundinarieae and 14 genera Bambuseae

and retrieved 6,089 GDs mapped at the MRCA of WBs, supporting a putative WGD event, namely here as Kappa (#8 in Fig. 1 and Supplementary Fig. 5), providing an opportunity to examine the genome evolution pattern of WBs and to compare with the previous models. We investigated Kappa further, as described below.

We performed phylogenomic analysis with multiple species (Analysis-I) for evidence supporting Kappa. Examination of the topology of gene trees with the GDs mapped at the MRCA of WBs revealed 3300 GDs with the (AB)(AB) type, 1515 the (AB)A type, and 1274 the (AB)B-type (Fig. 3a and Supplementary Fig. 31a). Here, A and B represent Bambuseae and Arundinarieae, respectively, and these GDs support kappa. We then examined syntenic pairs in genomes of *P. edulis*, *Dendrocalamus latiflorus*, and *Bonia amplexicaulis*[23,54,55], for correspondence to the GDs mapped at the MRCA of WBs (Supplementary Fig. 31a, b). In particular, among the GDs of the (AB)(AB) type, 91.0% of GDs in *D. latiflorus*, 84.5% of GDs in *B. amplexicaulis*, and 91.8% of GDs in *P. edulis* match the syntenic duplicates (Supplementary Fig. 31a), showing syntenic evidence for the kappa. In addition, for the GDs of the (AB)A type, syntenic duplicates from *D. latiflorus* and/or *B. amplexicaulis* match 918 GDs, of which 766 GDs corresponded to duplicates anchored in the synteny blocks that also contain gene pairs placed at the WB MRCA with the (AB)(AB) type (Fig. 3b and Supplementary Fig. 31a). Similarly, among the GDs of the (AB)B-type, 661 have syntenic genes from *P. edulis* and 634 of them are anchored in the synteny blocks where some genes match (AB)(AB)-type GDs mapped at the MRCA of WBs (Fig. 3b and Supplementary Fig. 31a). These results indicate that various duplicated genes in the syntenic blocks derived from the kappa contribute to different GDs of the (AB)(AB), (AB)A, and (AB)B types. Together, syntenic genes from the three WB genomes match 2213 GDs, ~36.3% of the 6,089 GDs, supporting the idea that the paleo-polyploidization, kappa, had preceded the divergence of the WB ancestor into Arundinarieae and Bambuseae.

The kappa event was also supported by a phylogenomic analysis with five sequenced genomes (Analysis-II). Analysis-I above focused on gene families that are shared by multiple species and hence might have missed some genes that are present in genome-sequenced species but not detected from the transcriptome datasets[1]. To examine the syntenic genes from those genomic blocks that contain gene pairs matching the kappa event, we analyzed another dataset consisting of five sequenced genomes (Supplementary Fig. 32a), which represent each of the Bambusoideae tribes and the other two subfamilies of the BOP clade. We identified 9,691 gene orthogroups with chromosomal collinearity and reconciled their gene trees with the species topology, placing 2430 GDs (BS ≥ 50) with nine topology types (T1 through T9 in Supplementary Fig. 32b) at the MRCA of WBs (Supplementary Fig. 32c). In addition, dot-plot of intraspecific chromosomal collinearity revealed that the syntenic blocks containing the gene pairs with T1 and/or T2 topologies occupy the most blocks (Fig. 3b and Supplementary Fig. 32d). Specifically, in the *P. edulis* genome, 91 syntenic blocks have gene pairs with the T1 and/or T2 topologies and a total of 12,268 syntenic gene pairs, which account for 71.5% of all detected syntenic gene pairs. Similarly, in the *D. latiflorus* genome, 220 syntenic blocks contain gene pairs with T1 and/or T2 topologies and a total of 20,580 gene pairs (accounting for 86.8% of 23,722 gene pairs). 1275 of 1288 GDs with the T3-T9 topologies are anchored in the syntenic blocks where some genes have the T1 and/or T2 topologies (Supplementary Fig. 32c, d), suggesting that these gene pairs with the T3-T9 topologies were also from kappa. Furthermore, a comparison of the syntenic paralogs of the T3-T9 topologies with the GDs mapped to the kappa using Analysis-I detected 206 shared GDs (Supplementary Fig. 32e). The difference in placements of some gene(s) in T3-T9 might be due to the small taxon number. Further branch length analyses of the two largest groups (T3-T4; Supplementary Fig. 32f, g) imply that some incorrect topologies might be due to relatively high substitution rates of some paralogues resulting in long-branch attraction (LBA)

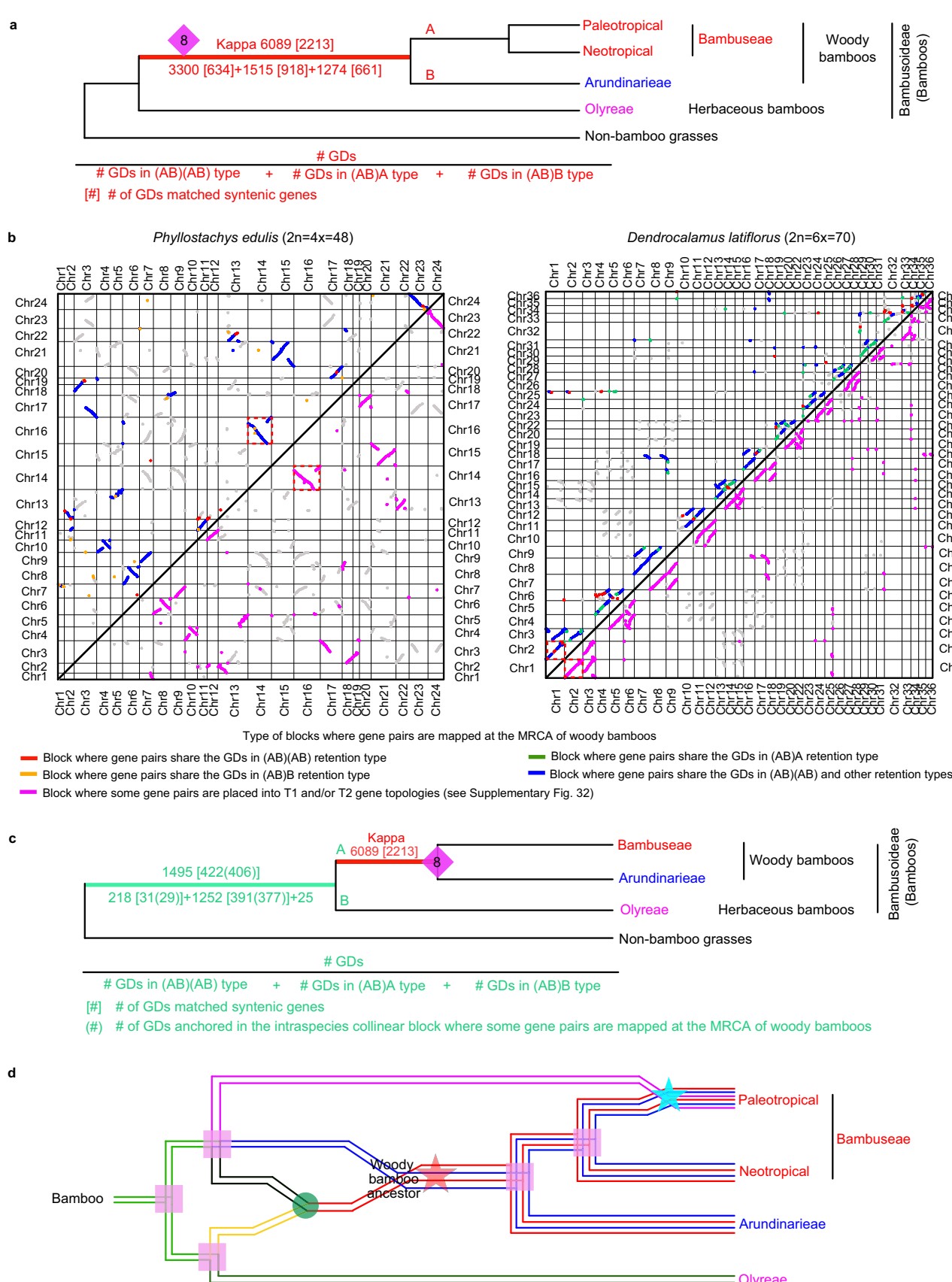

Type of blocks where gene pairs are mapped at the MRCA of woody bamboos

— Block where gene pairs share the GDs in (AB)(AB) retention type
— Block where gene pairs share the GDs in (AB)A retention type
— Block where gene pairs share the GDs in (AB)B retention type
— Block where gene pairs share the GDs in (AB)(AB) and other retention types
— Block where some gene pairs are placed into T1 and/or T2 gene topologies (see Supplementary Fig. 32)

**Fig. 3 | Identification of the WGD shared by woody bamboos. a** Number of GDs (red numbers) shared by woody bamboos (WBs) (see also Supplementary Fig. 5). Number in square brackets represents the number of GDs matched by syntenic genes. See Supplementary Fig. 31 for the number of GDs (and GDs matched by syntenic genes) shared by individual species in distinct retention types. #8 in purple rhombus is as in Fig.1 and named here as Kappa. The branches of Bambuseae and Arundinarieae, respectively, are labeled with capital letters A and B, for description of retention type. **b** Dot-plots illustrating intraspecific collinear blocks (continuous points) of *Phyllostachys edulis* (left) and *Dendrocalamus latiflorus* (right) genomes ordered by chromosomes. In each dot-plot, red, blue, green, or orange dots in the top-left part show the GDs mapped at the MRCA of WBs in Supplementary Fig. 5, and the pink dots in the bottom-right part shows the GDs mapped at the MRCA of WBs in Supplementary Fig. 32; different colors represent different retention types as shown below the dot-plots. Gray blocks represent syntenic genes not included in

gene trees due to the lack of sufficient species with the gene or low BS support or mapped to other positions. **c** Number of GDs (green numbers) mapped at Bambusoideae. The meaning of numbers in square brackets is as in Fig. 3a and also shown below. The number in round parentheses represents the number of GDs matched by syntenic genes anchored in the syntenic blocks with some other gene pairs mapped at the WB MRCA. Capital A, WBs; Capital B, herbaceous bamboos (HBs). See details of retention type in Supplementary Fig. 33. **d** A model of woody bamboo genome evolution. Red star, the Kappa event. Green circle, hybridization between WB ancestor and HBs. Pink squares, speciation events. Blue star, the WGD event shared by hexaploids in Bambuseae. The parental lineage of the third subgenome of paleotropical bamboos in Bambuseae is proposed to have originated in parallel to the two parental lineages of WB ancestor (see topologies in Supplementary Fig. 37). Source data are provided in a Source Data file.

artifacts[56]. In short, our phylogenomic study using the five-genome dataset placed a large number of syntenic GDs supporting the kappa event at the MRCA of WBs.

Further support for kappa was detected from Ks analysis (Analysis-III). Ks analysis of syntenic genes from the *P. edulis* genome revealed a recent peak at 0.186 (Supplementary Fig. 20b), corresponding to a previously identified paleo-polyploidization event in this genome[22]. Ks analyses suggested that this paleo-polyploidization in *P. edulis* (Arundinarieae) was also shared by *B. amplexicaulis* (Bambuseae) but not by *L. pauciflora* (Olyreae) (Supplementary Fig. 20b). In addition, a comparison of 134 syntenic blocks with Ks values of 0.0−0.46 and 111 syntenic blocks containing gene pairs mapped at the MRCA of WBs in Analysis-I reveals 106 shared blocks, whose syntenic genes account for 97.3% of the gene pairs in the 134 syntenic blocks in Analysis-III and 99.1% of gene pairs in the 111 blocks for Analysis-I (Supplementary Fig. 20b). Therefore, the paleo-polyploidization event supported by synteny in *P. edulis* genome is the kappa event. Molecular dating analysis estimated the kappa event at ~62.05 Ma (million years ago), which was earlier than the divergence time (55.67 Ma) between Arundinarieae and Bambuseae and later than that (69.79 Ma) between Arundinarieae and Olyreae (Fig. 1 and Supplementary Data 3).

## Possible origin of Kappa and the Kappa-derived genes retained at tribes

Besides the GDs at the MRCA of WBs supporting the kappa event, our Analysis-I also retrieved 1495 GDs at the MRCA of Bambusoideae (Fig. 3c and Supplementary Fig. 5). To assess whether the GD cluster is from a WGD before the Bambusoideae divergence, we classified GDs into different retention types and identified 83.7% (1252) of GDs with the topology of [(WB, HB)WB] ((AB)A; Fig. 3c and Supplementary Fig. 33), suggesting that only one copy of the gene pairs was shared by Bambusoideae. Examination of bamboo gene positions indicates that 406 GDs correspond to the duplicates in the synteny blocks that also have some genes mapped at the WB MRCA (Fig. 3c and Supplementary Fig. 33). This result suggests that the WB paralogues corresponding to GDs mapped at Bambusoideae might also be from kappa and that the WB ancestor might be from a hybridization (or possibly introgressions; for convenience "hybridization" will be used hereafter) involving HBs as a parental lineage.

Possible progenitors of Kappa were inferred. The GD cluster mapped at Bambusoideae could be from kappa but were incorrectly placed on Bambusoideae, possibly due to LBA artifacts. To test this possibility, we reconstructed gene trees using the first and second codon positions, which are less prone to LBA, for 242 (WB, HB)WB-type GDs with non-bamboo grasses as outgroups in Analysis-I (Supplementary Fig. 34a, b). The topologies of the re-generated gene trees placed 175 GDs (BS ≥ 50) at Bambusoideae (Supplementary Fig. 34b), consistent with the GD burst at Bambusoideae from the above-mentioned analyses using full-codons. In addition, HB genes have a significantly longer branch length than WB genes in both trees from

full codons and the 1st+2nd codon positions (Supplementary Fig. 34c), likely owing to a shorter lifecycle and more mutations of HBs than WBs[53,57]. These results suggest that long-branch attraction could not fully explain the detected GDs at the MRCA of Bambusoideae.

Therefore, an ancient hybridization between an HB-related lineage and another diploid parent (possibly extinct) could be the polyploid origin of WBs and provide an explanation for the GDs at Bambusoideae (Fig. 3d). Thus, some WB genes might be derived from the (putative) HB-related parent with syntenic support, including single-copy WB genes. To examine the relationship between the single-copy WB genes and their HB homologues, we compared two WB genomes (*P. edulis* and *D. latiflorus*) with the herbaceous *Olyra latifolia* genome[23,54,55] and detected syntenic blocks as possibly from hybridization (Supplementary Figs. 35, 36), supporting the idea that single-copy WB genes were derived from the progenitor related to HB (*O. latifolia*). To further investigate the hybridization model (Fig. 3d), we examined the topologies of gene trees generated in the Analysis-II (Supplementary Fig. 32) for a duplication mapped at Bambusoideae and retrieved 631 gene trees (including 296 gene trees with ≥50% BS; Supplementary Fig. 37a) that placed HB as sister to a WB clade, consistent with the above hybridization model. Specifically, genes related to hybridization were enriched in the Brassinosteroid metabolism (GO:0016131; Supplementary Fig. 38a), such as Brassinosteroid-dependent 1 (*BRD1*) gene with roles in shoot internode elongation in maize and rice[58,59], with non-HB-related WB homolog of *BRD1* in *P. edulis* implicated in the rapid growing of WBs (Supplementary Fig. 38b–d).

GDs mapped to woody bamboo tribes were likely from kappa. The Analysis-I placed 1688 GDs at Arundinarieae, with 1496, 125, and 67 GDs being the (AB)(AB), (AB)A, and (AB)B types, respectively (Fig. 4a and Supplementary Fig. 39), but it was not clear these GDs were from kappa or another WGD. We identified 853 (AB)(AB)-type GDs corresponding to gene pairs in *P. edulis*, including 800 in syntenic pairs (Supplementary Fig. 39a, b). Approximate 97.9% of the syntenic pairs are anchored in the synteny blocks that also include paralogues matching the GDs at the MRCA of WBs, implying that these GDs (at Arundinarieae) are also from kappa (see an example in Fig. 4b). Similarly, our phylogenomic analyses also placed 444 GDs at Bambuseae (Fig. 4c and Supplementary Fig. 40), with 206 correspond to syntenic genes in *B. amplexicaulis* and *D. latiflorus*; 181 of these 206 GDs are anchored in the syntenic blocks that also contain duplicates placed at WB ancestor (Fig. 4c, d and Supplementary Fig. 40a), again suggesting these GDs retained at Bambuseae were from kappa.

We further investigated the third subgenome of the hexaploid tropical woody bamboos. According to the previous ABCD genomic model of WBs, the A subgenome is more similar to the B subgenome instead of the C subgenome shared by WBs[23,53]. To probe the position of the third subgenome of hexaploid bamboos using phylogenomic analyses, we selected among the 9,691 gene trees in Analysis-II those with three gene copies from the hexaploid *D. latiflorus* genome and

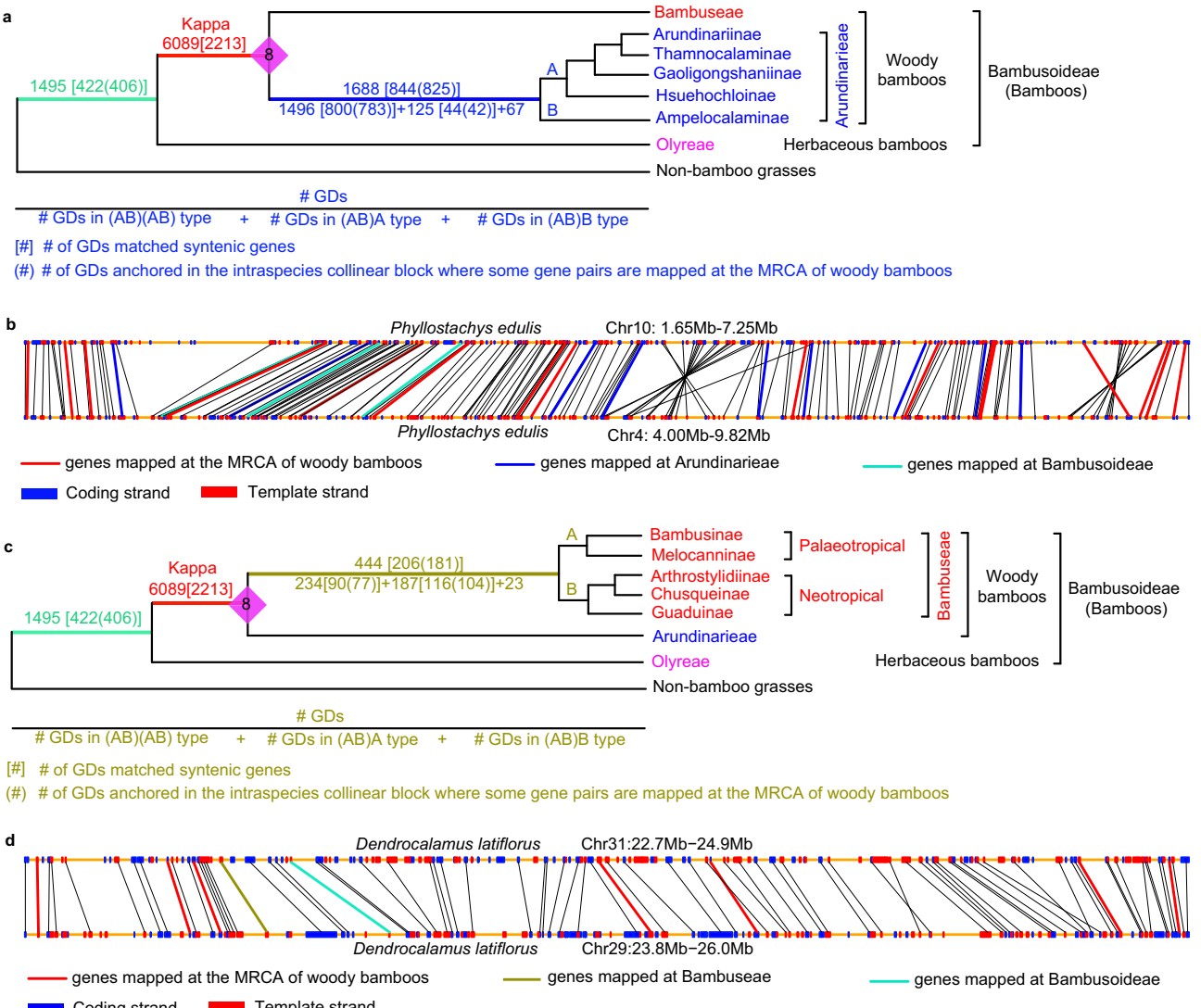

**Fig. 4 | GDs mapped at Arundinarieae and Bambuseae and their relationship to kappa. a** Number of GDs mapped at Arundinarieae. See Supplementary Fig. 39 for the number of GDs (and GDs anchored in genome collinear blocks) shared by different species in Arundinarieae. The meaning of numbers in square brackets and round parentheses is as in Fig. 3 and, also shown below. Capital A, the MRCA of Arundinariinae and Hsuehochloinae subtribes; capital B, Ampelocalaminae subtribe. **b** Illustration of an intraspecific collinear block of *Phyllostachys edulis*. Lines between different chromosome fragments represent collinear genes, some of which are linked by thick and colored lines indicating different phylogenetic positions of the duplication shared by the genes as shown below the blocks. The meaning of red and blue rectangles is the same as that in Fig. 2b. **c** Number of GDs mapped at Bambuseae. See Supplementary Fig. 40 for the number of GDs (and GDs anchored in genome collinear blocks) shared by different species in Bambuseae. The meaning of numbers in square brackets and round parentheses is as in Fig. 3 and, also shown below. Capital A, the palaeotropical Bambuseae; capital B, the neotropical Bambuseae. **d** Illustration of an intraspecific collinear block of *Dendrocalamus latiflorus*. The meaning of lines between different chromosomal fragments is the same as that in part **b**. The meaning of red and blue rectangles is the same as in Fig. 2b. Source data are provided in a Source Data file.

two genes from *P. edulis*. Among 1271 gene trees with the MRCA of three *D. latiflorus* genes (BS ≥ 50) being WBs (Supplementary Fig. 37b), 440 (T1, Supplementary Fig. 37b) have two WB copies (each from both *P. edulis* and *D. latiflorus*), with the third copy of *D. latiflorus* genes sister to the combined clade with both WB copies, providing evidence for the possible origin of the third hexaploid subgenome being a diploid related to the WB ancestor (Fig. 3d).

**Ancient hybridization contributing to the diverse adaptation of a tetraploid wild rice**

Oryzoideae species usually grow in wetlands, rivers, seasides, and forests[14,60]. Among Oryzoideae genera, *Oryza* includes two domesticated rice species *O. sativa* and *O. glaberrima*, and has several sequenced genomes, with six diploids (A, B, C, E, F, G; 2n = 2x = 24) and five allotetraploids (2n = 4x = 48)[27]. For example, *Oryza coarctata* is a recently sequenced tetrapolyploid wild rice[33] (KL genome and formerly HK genome[61,62]); it is a halophytic (salt-tolerant) plant distributed in coastal regions and can be submerged under seawater repeatedly[60]. *Oryza barthii* (A-genome), *O. punctata* (B genome), *O. officinalis* (C genome), and *O. australiensis* (E genome) are exposed to seasonally dry environments, whereas *O. brachyantha* (F-genome) usually grows in rocky tidal pools[14]. Hence, the *Oryza* genus provides great resources to study genomic adaptation to variations in water availability, with candidate genes related to the submergence in rice (*O. sativa*) (e.g., *SUB1*[63]) and a deep-water variety of rice (e.g., *SD1*[41]). Transcriptome sequencing in *O. coarctata* has linked NAC, MYB, and WRKY putative transcription factors to salinity tolerance and implicated the bZIP, bHLH, HSF, and AP2-EREBP family members in submergence stress

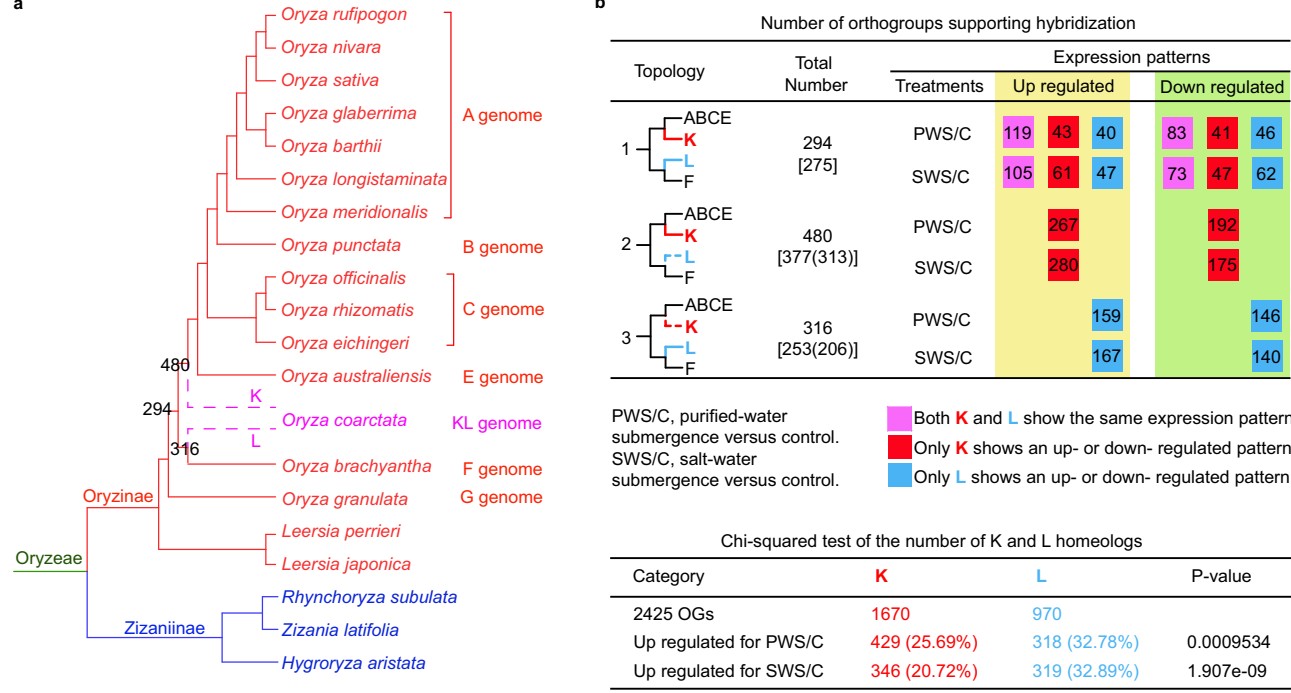

**Fig. 5 | Hybridization in Oryzoideae. a** An Oryzeae phylogeny with an emphasis on *Oryza* genomes and a model of parental lineages of the tetrapolyploid wild rice *O. coarctata*. Six capital letters (A, B, C, E, F, G) to the right of species names indicate diploid *Oryza* genome types and KL indicates the tetrapolyploid type of *O. coarctata*. The dotted lines represent two putative progenitors of the KL genome, with one parental lineage (referred to as K subgenome) sister to the MRCA of A, B, C, E genomes and the other one (referred to as L subgenome) sister to F genome. Support of the *Oryza* phylogeny from different datasets can be found in Supplementary Figs. 41–43 and Supplementary Data 4. See part **b** for the meaning of three numbers (294, 480, and 316). **b** The upper table shows expression patterns of genes from different subgenomes K and L. Topology-1, K as sister to the MRCA of A, B, C, and E genomes (ABCE), and L as sister to the F genome. Topology-2, K as sister to the ABCE MRCA, but L was lost. Topology-3, K was lost and L as sister to the F

genome. The number of orthogroups matching the topology is shown to the right of the topology. The number in square brackets represents the number of homeologues matched by syntenic genes (see an example in Supplementary Fig. 44). Number in round parentheses for Topology-2 or 3 represents the number of homeologues corresponding to syntenic genes that are anchored in interspecific blocks with some genes matching Topology-1. Number in colored squares represent different number of orthogroups with respective expression patterns of K and L genes as shown below the table. See Supplementary Data 4 for K and L homeologs with their gene annotations and expression values. The lower table shows a summary of the number of K and L homeologues for different expression patterns, with a one-sided Chi-squared test. The number in round parentheses represents the percentage of the upregulated orthogroups in their respective total orthogroups. Source data are provided in a Source Data file.

response[64]. The K genome of *O. coarctata* was proposed to be related to the ABC genomes using the *Adh2* gene phylogeny, but the origin of the L(H) genome was not clear[60,61,65]. The uncertainty of both the phylogenetic position and progenitors of the K and L genomes has hindered the understanding of the possible effect of allo-tetraploidization of *O. coarctata* on the evolution of gene functions related to submergence tolerance. The published *Oryza* genomes (and pan-genomes), including a scaffold-level assembly of the *O. coarctata* genome[28–33], together with our phylogenomic datasets, allowed an investigation into the placement of the K and L genomes and the possible contribution of KL-related homeologues to adaptations to diverse environments.

Possible progenitors of the KL genomes were investigated by placing *O. coarctata* in the *Oryza*/Oryzoideae phylogeny. We utilized ASTRAL-Pro to infer the phylogenetic relationships among *Oryza* species, with monophyletic *Oryza* as sister to *Leersia* (Supplementary Fig. 41). In particular, *O. coarctata* is sister to the MRCA of species with A, B, C, and E genomes (referred to as the ABCE ancestor hereafter) (Supplementary Fig. 41). Furthermore, among 10,615 gene trees that contained at least one *O. coarctata* gene, 3555 support (≥50 BS) the sisterhood of *O. coarctata* and the ABCE ancestor and 3208 support *O. coarctata* being sister to the F-genome lineage (Supplementary Fig. 42). These results suggest that the progenitors of *O. coarctata* might be related to taxa containing the ABCE ancestor and the F genome, respectively (Fig. 5a). To further test this hypothesis, we selected 2425 gene trees containing at least one sisterhood between *O.*

*coarctata* and the ABCE ancestor or at least one sisterhood between *O. coarctata* and the F-genome lineage and pruned the gene trees by removing putative paralogues from duplications before the *Oryza* diversification (see methods). We also removed 21 orthogroups with low-quality alignments and reconstructed 2404 gene trees. In 2404 pruned gene trees (orthogroups), the *O. coarctata* gene sister to the ABCE ancestor is designated as 'K' and the gene sister to the F-genome as 'L', as proposed previously[61,62]. ASTRAL analysis of the 2404 trees revealed support for the K genome of *O. coarctata* sister to the clade with A, B, C, and E genomes (BS = 100%; Supplementary Fig. 43) and for the L genome being sister to the F genome lineage (BS = 100%; Supplementary Fig. 43). Further ASTRAL analysis of 2398 gene trees that show genomic collinearity for *Oryza* species (Supplementary Fig. 43) provides genomic support for a model of K and L genomes being related to the ABCE ancestor and the F-genome, respectively (Fig. 5a).

Among the 2404 gene trees with *O. coarctata* genes, 1090 had topologies that support hybridization for *O. coarctata*, with G genome or other Oryzinae species as outgroup (Fig. 5b). These included 294 orthogroups with Topology-1 [(K, ABCE) (L, F)] and both clades having BS ≥ 50% (Fig. 5a, b), and 275 (93.5%) of them match syntenic genes from *O. coarctata* and/or interspecific collinear genes between *O. coarctata* and diploid *Oryza* genomes (see Supplementary Fig. 44 for example). These syntenic blocks thus likely contain other K and L homeologues. In addition, 480 orthogroups with topology-2 [(K, ABCE) (F)] could have lost the L paralogue and 316 orthogroups with topology-3 [(ABCE) (L, F)] might have retained the L and but lost the K

paralog (Fig. 5a, b), providing additional evidence for the origin of *O. coarctata* being a hybrid of K and L genomes. Furthermore, 377 orthogroups with topology-2 also include the *Oryza* collinear genes as described above; among the collinear genes in these orthogroups, 313 (83%) are anchored in the same blocks that also include collinear genes corresponding to the topology-1 (Fig. 5b). Similarly, 253 orthogroups with topology-3 include the *Oryza* collinear genes, and the collinear genes in 206 (81.4%) orthogroups are anchored in the blocks containing some collinear genes exhibiting topology-1 (Fig. 5b). The presence in the same syntenic blocks of genes from orthogroups with topology-1 and topology-2, or from orthogroups with topology-1 and topology-3, further supports the idea that the genes with all three topologies were derived from the same ancestral K and L genomes.

Furthermore, we tested whether the K and L homeologues might have contributed to the adaptation of *O. coarctata*. Specifically, we focused on possible adaptation to high salinity and submergence, in part because their putative parental lineages are related to extant species with distinct habitats. For example, differential gene expression in response to submergence in salt water could suggest a possible role in such adaptation. Indeed such expression differences for the 1090 orthogroups (Fig. 5b) were detected using published transcriptome data[64]. Specifically, expression levels of K and L homeologues were compared for plants under purified-water submergence (fully submerged in reverse osmosis water), salt-water submergence (completely submerged in 450 NaCl solution) or control conditions (Supplementary Data 4). Among the K and L homeologues with topology-1, 202 pairs show consistent expression patterns under purified-water submergence versus control (PWS/C) for both homeologues (119 both up and 83 both down; Fig. 5b pink squares). In addition, both K and L homeologues in 105 orthogroups with topology-1 were upregulated under salt-water submergence versus control (SWS/C), and both downregulated in 73 orthogroups (Fig. 5b pink squares). When considering both submergences, 82 orthogroups had both K and L homeologues being upregulated under both PWS/C and SWS/C, and 58 orthogroups had both homeologues downregulated under both conditions. Seven other pairs of homeologues show upregulation only under one submergence but down in the other. The differential expression of these 147 (= 82 + 58 + 7; Supplementary Data 4) pairs of homeologues suggests that genes from both subgenomes might have contributed to the adaptation to submergence in purified water and/or salt water. 147 other homeologs of the topology-1 show different expression patterns (Supplementary Data 4); for example, 43 homeologs with K upregulated under PWS/C, and 61 homeologs with K upregulated under SWS/C (Fig. 5b red squares). Additionally, orthogroups with topologies 2 and 3 contain only K or L homeologues, respectively. For the K genes (topology-2), the numbers of differentially expressed genes under PWS/C (459 = 267 + 192; Fig. 5b) or SWS/C (455 = 280 + 175) are similar. Also, for the L genes, the number of differential expressed genes under PWS/C (305 = 159 + 146) is almost the same as that for SWS/C (307 = 167 + 140). Combining orthogroups of all three topologies, a greater number of K homeologues were upregulated under PWS/C than SWS/C (Fig. 5b, lower table); on the other hand, greater percentages of L homeologs were upregulated under both submergences than K homeologues.

Among the differentially expressed K and L homeologues under PWS/C and SWS/C, several encode putative transcription factors (e.g., NAC, MYB, ERF, WRKY, and bZIP; Supplementary Data 4). Other differentially expressed homeologues encode transporters (e.g., potassium transporter), transferases (e.g., spermidine synthase), oxidoreductase (e.g., superoxide dismutase), and others (Supplementary Data 4). In addition, the hypothesis that two genes of topology-1 for the vacuolar H⁺-ATPase subunit B (*VHA-B*; *PC_40923* and *PC_47707* in Supplementary Data 4) play a role in vacuolar $Na^+$ storage and salt tolerance is supported by the enhanced *Arabidopsis* salt

tolerance due to overexpression of the *VHA-B* gene from halophyte *Halostachys caspica*[66]. Similarly, among topology-2 genes, *O. coarctata* gene (*PC_29650* in Supplementary Data 4) is homologous to the rice *WRKY71* gene (*Os02g0181300 = LOC_Os02g08440*), whose expression is related to salt tolerance in rice[67]. For topology-3 genes, one example is *SOD4* for a superoxide dismutase 4 (*PC_11655* in Supplementary Data 4), which was reported to play a protective role against damage from salt stress[68]. The evolutionary and differential expression patterns support functional hypotheses to be tested with further analyses and can facilitate rice genetics and breeding of tolerance to salt stress and submergence stress.

### Retention of ancestral tandem duplications in Oryzoideae

In the subfamily Oryzoideae, our above phylogenomic analyses support a WGD shared by *Zizania latifolia* and *Rhynchoryza subulata* in the subtribe Zizaniinae of the tribe Oryzeae (#7; Fig. 1 and Supplementary Figs. 6, 20a); two other clusters of 296 GDs (#29; Supplementary Fig. 6) and 237 GDs (#30; Supplementary Fig. 6) were, respectively, mapped at Oryzinae and Oryzeae and contain tandem duplications (TDs), which were previously defined as paralogues with fewer than four intervening genes[69]. TDs and other recent gene duplications (e.g., transposition-related duplicates) in rice were thought to be largely species-specific (954 gene families retaining the rice-specific tandem duplications)[70]. However, their analyses[70] did not include genomes of other close relatives of rice. Examinations of evolutionary patterns of tandem duplications require well-assembled genomes with nearly complete gene annotations, but have not been reported for rice and its close relatives. Here the 296 GDs mapped at Oryzinae were further examined for tandem duplication patterns using 16 sequenced genomes (Figs. 5a, 6a; all 15 *Oryza* species and *Leersia perrieri*), covering six diploid genome types in *Oryza*. We detected tandem paralogues supporting 193 duplications (Fig. 6a). In addition, different species has retained different tandem duplicates (for instance, duplicates for 104 GDs in *O. barthii*, 93 GDs in *O. sativa*, 52 GDs in *O. australiensis*, 44 GDs in *O. brachyantha*, and 89 GDs in *L. perrieri*; Supplementary Fig. 45). It is possible that the tandem duplicates have contributed to the divergence in Oryzinae. Similar to the retention of ancestral tandem duplications mapped at Oryzinae, estimations of the gene order of duplicates mapped at Oryzeae also support that 148 of 237 GDs were from tandem duplications (Fig. 6a and Supplementary Fig. 46).

Among the gene trees that show tandem duplications at Oryzinae and/or at Oryzeae, 161 gene trees contain tandem duplications at Oryzinae, whereas 116 contain tandem duplications at Oryzeae, including 16 gene trees containing tandem duplications at both Oryzinae and Oryzeae. To illustrate the specific gene duplication history, we describe the genes encoding peroxidases (Fig. 6b). Peroxidase gene tree and genomic positions suggest that two copies (#1 and #2) were produced through a tandem duplication before the split between Oryzinae and Zizaniinae and after the divergence of Oryzeae from Ehrharteae, another tribe of Oryzoideae. In addition, copy #2 experienced tandem duplication again in Oryzinae to generate #2a and #2b paralogs (Supplementary Fig. 47). In some *Oryza* species with the A-genome, copy #1 also further doubled through an A-genome-specific tandem duplication (Supplementary Fig. 47) and the resulting paralogues include two rice genes (1a: *LOC_Os06g32960*; 1b: *LOC_Os06g32980*) among six rice peroxidase genes; furthermore, three genes (#2b-1: *LOC_Os06g33080*; #2b-2: *LOC_Os06g33100*; and #2b-3: *LOC_Os06g33090*) in the #2b clade were produced by two rice-specific tandem duplications (Fig. 6c and Supplementary Fig. 47). In addition, the expression of all six rice peroxidase genes (Fig. 6c) was upregulated in roots under submergence[71], suggesting a positive role of these tandem duplicates for response to submergence stress. Peroxidase genes belong to a superfamily in plants and play key roles in reactive oxygen species (ROS) formation[72]. ROS are signaling molecules that regulate

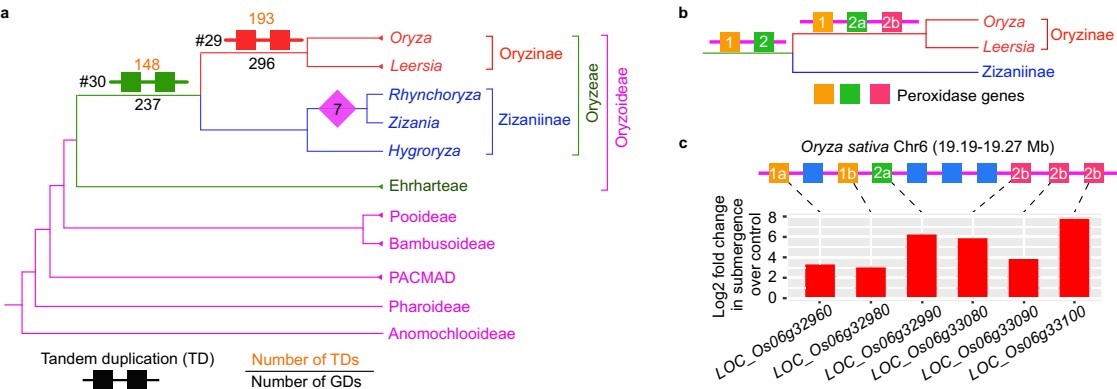

**Fig. 6 | Tandem duplications in Oryzoideae. a** Tandem duplications (TDs) in Oryzeae. The cladogram depicts the generic relationships in Oryzeae simplified from Supplementary Fig. 41. Number above the branch indicates the number of TDs. See Supplementary Figs. 45–46 for details of TDs mapped at Oryzinae (#29) and at Oryzeae (#30), respectively. #7 indicates a WGD event shared by *Zizania* and *Rhynchoryza* (see Supplementary Fig. 20a for more information). **b** Illustration of the duplication of peroxidase genes successively from the MRCA of Oryzinae and Zizaniinae to Oryzinae. Numbered squares in different colors represent different

ancestral and intermediate copies of the peroxidase genes inferred from the phylogenetic and genomic results shown in Supplementary Fig. 47. **c** The order of 6 peroxidase genes in the *Oryza sativa* chromosome 6 and their expression patterns. The meaning of numbered squares is the same as that in part b. Different recent tandem duplicates with the same number and lowercase letters (such as 2b) were derived from the same ancestral gene indicated in part b. A phylogenetic tree of the six rice genes is shown in Supplementary Fig. 47. Blue squares represent non-peroxidase genes. Source data are provided in a Source Data file.

response to submergence and other oxidative stresses[73]. Together, our results suggest that increasing copy number of peroxidase genes via successive tandem duplications at several points in Oryzeae history likely enhanced evolutionary adaptation to water environments.

## Differential retention and loss of rho-derived duplicates and potential functional consequences

Following a WGD, differential retention and loss of gene duplicates are considered important for functional divergence among descendant lineages[35]; however, this problem has not been analyzed in detail in a plant family previously due to insufficient sequenced genomes. The Poaceae rho event and available sequenced grass genomes of multiple subfamilies, combined with a robust Poaceae phylogeny[13], provide a great opportunity to investigate evolutionary patterns of differential losses of duplicates among subfamilies and their potential functional implications. For ease of description and discussion here, a grass (gene) orthogroup is defined as those genes that descended from a single ancestral gene after Poaceae diverged from other Poales families but before rho, including both orthologues and paralogues due to Poaceae-specific duplications. The above phylogenomic analyses of grass orthogroups showed that different numbers of gene pairs from rho were retained in different species; for example, 840 rho-derived gene pairs were retained in sorghum, 847 such gene pairs in rice, and 1018 pairs in *Pharus latifolius* (Supplementary Fig. 26a).

We investigated differential retention/loss patterns of rho-derived gene pairs for Poaceae subgroups. First, we detected orthogroups supported by syntenic genes from 24 sequenced grass genomes, with pineapple as an outgroup species in Poales (Supplementary Fig. 48a, b). To obtain further support for rho-derived duplicates, we integrated our phylogenomic results (Supplementary Fig. 3) and the synteny results by identifying the synteny blocks that contains at least one gene pair belonging to an orthogroup with a GD mapped at the MRCA of Poaceae or one of the early nodes with multiple subfamilies (C1–C3 in Fig. 2a). To illustrate this analysis, an example of synteny blocks is shown in Fig. 7a (see details in Supplementary Fig. 49a), the #6 genes correspond to a GD mapped at Poaceae in the gene tree (Supplementary Fig. 49b), supporting the gene pairs (pink) in the syntenic block being from rho. The gene trees of the orthogroups were reconciled with species-tree to estimate the retention and loss events after rho in different subfamilies (see methods), and the results

revealed that 6147 orthogroups retained a single copy at Poaceae and 2758 orthogroups retained in pair (Fig. 7b, Supplementary Fig. 48b, and Supplementary Data 5). Among the 6147 orthogroups, 5666 experienced subsequent lineage-specific duplication in at least one subfamily (Type-I; Fig. 7c and Supplementary Data 5); one such orthogroup contains the fertilization-independent endosperm (*FIE*) genes[74] with a duplication in Panicoideae (e.g., maize *FIE* genes [*Zm00001d049608*(*FIE1*) and *Zm00001d024698*(*FIE2*)]; Supplementary Fig. 50). Other instances of Type-I orthogroups include the *TASSELSEED2* (*TS2*)[75], *DWARF53* (*D53*)[76], *COLD1*[77], and *NAC78*[78] (see details of examples in Table 1). For the remaining 481 of the 6147 orthogroups, no more than one copy was detected in grass species (Type-II; see orthogroups in Supplementary Data 5 and a specific gene tree in Supplementary Fig. 51). Among 2758 orthogroups with two rho-derived copies (Fig. 7c and Supplementary Data 5), 128 (Type-III) have two copies in each of the four subfamilies (Supplementary Fig. 52); whereas 2630 (Type-IV) have lost one or two detected copies in at least one subfamily (see gene examples in Table 1). Among the Type-IV orthogroups, we identified four patterns (IV-1 through IV-4 in Fig. 7d). Specifically, 1991 have two or more subfamilies with one detected copy (IV-1 and IV-2), including 578 that exhibit reciprocal loss of paralogues in different subfamilies (IV-2; see an example in Supplementary Fig. 53). Among the 2758 orthogroups with two copies in at least one subfamily, 565 show possible reciprocal loss of rho-derived duplicates between species within an individual subfamily (Supplementary Data 5). These detected patterns should be further tested by including more high-quality genomes from different subfamilies of Poaceae and other families of Poales.

To gain clues regarding possible functions of these 8905 (= 6147 + 2758) orthogroups, we analyzed GO terms (see details in methods). Among the GO terms enriched significantly in three divergent genomes (rice, barley or wheat, and maize or sorghum), they were classified into several broad categories (Supplementary Data 6), including regulation of gene expression (nucleic acid binding), protein regulation and modification (enzyme binding), (unspecified) metabolism (e.g., oxidoreductase activity and lyase activity), small-molecule metabolism (e.g., lipid binding and carbohydrate-binding), interaction of proteins (e.g., calmodulin binding), and nucleic acid metabolism (helicase activity). The results imply that the ancestral genes from rho might play important roles in the regulation of RNA and protein synthesis and metabolism. We also compared the GO annotations for

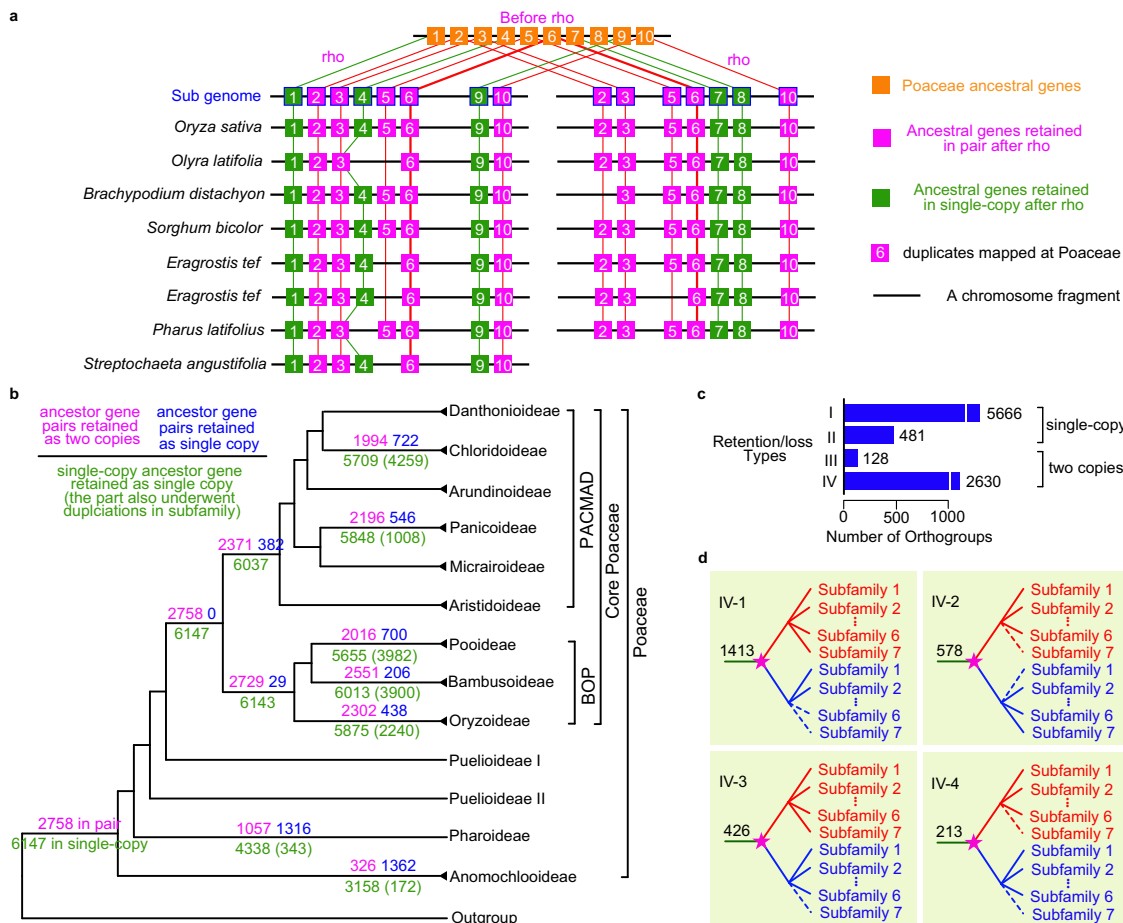

**Fig. 7 | Retention and losses of ancestral genes from rho. a** Illustration of an interspecific collinear block among grasses used for inferring ancestral genes. Numbers in squares represent different genes; the same number among species indicates the same orthogroup. Pink and green squares indicate the retention in pairs and in single-copy, respectively. Orange squares indicate the inferred ancestral genes before rho. The orthogroup with thick lines indicates duplicates phylogenetically mapped at Poaceae (see details in Supplementary Fig. 49). **b** Number of ancestral genes from rho in pair and in single-copy among grasses. The cladogram depicts subfamilial relationships of grasses simplified from the tree in Supplementary Fig. 48. Pink numbers, the number of ancestral gene pairs retained as two copies; blue numbers, the number of ancestral gene pairs retained as single copy; and green numbers, the number of single-copy ancestral gene retained. **c** Bar plots showing the number of the rho-derived orthogroups with four different retention/loss patterns. Type-I represents the single-copy orthogroups with further lineage-specific duplications in at least one subfamily. Type-II represents the single-copy

orthogroups with no more than one gene being retained in each species. Type-III represents the orthogroups with two copies being retained in each subfamily. Type-IV represents the orthogroups with two copies being retained in at least one subfamily and losses of detected copies in at least one subfamily. Sample gene trees of the types I through IV can be found in Supplementary Figs. 50–53. Numbers on the right of the bars represent the number of orthogroups. See Supplementary Data 5 for the retention number in each subfamily of these orthogroups. **d** Illustration of four schematic retention/loss patterns of the orthogroups in type-IV. Number on the branch represents the number of orthogroups (See Supplementary Data 5). Type-IV-1 represents a different number of retained subfamilies between the rho-derived paralogues. Type-IV-2 represents reciprocal losses of subfamilies between the rho-derived duplicates. Type-IV-3 represents losses of one detected copy in one subfamily. Type-IV-4 represents losses of the rho-derived paralogs in one or more subfamilies. Source data are provided in a Source Data file.

orthogroups in four different types and found that nine GO terms were exclusively detected in Type-I (such as basal transcription machinery binding, lipid transporter activity, and signaling receptor binding) (Supplementary Data 6). On the other hand, 13 GO terms were commonly detected in all four types, such as nucleic acid binding and DNA-binding transcription factor activity for the regulation of gene expression, protein dimerization activity in the category of 'Interaction of proteins', transferase/oxidoreductase/hydrolase/ligase activity in the metabolism category, catalytic activity acting on a protein, and ion binding for small-molecule metabolism (Supplementary Data 6).

We examined a specific pattern of differential retention/loss called the P̲air R̲etained in O̲ne lineage but S̲ingle-copy in O̲ther L̲ineages (PROSOL). These types of genes might have contributed to lineage-specific functions and adaptation. For example, phylogenetic analysis of the rice *MYB35* and *MYB36* genes important for

reproductive development and their homologs in other grasses indicated that they belong to an orthogroup with two rho-derived copies, one retained in the core Poaceae (*MYB35*, lost in the early-divergent subfamilies) and the other in Poaceae (*MYB36*, also present in Anomochlooideae), with the pineapple *MYB35/36* homologue as the outgroup[19]. However, the landscape of PROSOL genes in grasses and their functional implications remain largely unknown.

Here, we focused on four large subfamilies, Panicoideae, Pooideae, Bambusoideae, and Oryzoideae, that have economically important species and more available sequenced genomes and exhibit different environmental adaptations and morphologies. For convenience, we refer to a PROSOL specific to each of Panicoideae, Pooideae, Bambusoideae, and Oryzoideae, respectively, as PROSOL-Pa, PROSOL-Po, PROSOL-Ba, and PROSOL-Or. Our examination of the above-mentioned 2630 orthogroups in Type-IV uncovered 19 PROSOL-Pas, 8 PROSOL-Pos, 36 PROSOL-Bas, and 18 PROSOL-Ors, possibly

**Table 1 | Representatives of the rho-derived genes with different retention and loss patterns**

| Type | Orthogroup | Gene | Representatives | Function |
|---|---|---|---|---|
| I | HOG00220 | *DWARF8 /Rht-B1b* | *Zm00001d033680, TraesCS4B01G043100* | The maize *DWARF8* gene and the wheat *Rht-B1b* gene regulate GA dose-response[174]. |
| I | HOG00224 | *TT3.2* | *LOC_Os03g49940* | OsTT3.2 interacts with OsTT3.1 can enhance rice thermotolerance and reduce grain-yield losses caused by heat stress[175]. |
| I | HOG01525 | *NAC78* | *Zm00001d027395* | *ZmNAC78* can modulate the mRNA abundance of Fe transporters in kernels[78]. |
| I | HOG01763 | *COLD1* | *LOC_Os04g51180* | *OsCOLD1* interacts with G protein to confer chilling tolerance in rice[77]. |
| I | HOG05586 | *FIE* | *Zm00001d049608, Zm00001d024698* | *FIE* represses the endosperm development without fertilization[176]. |
| I | HOG05693 | *D53* | *LOC_Os12g01360* | *D53* represses the strigolactone signaling in rice[76]. |
| I | HOG07259 | *TS2* | *Zm00001d028806* | *TS2* encodes a short-chain alcohol dehydrogenase and has general developmental roles[75]. |
| II | HOG01539 | *TMT3B* | *TraesCS4B01G322000* | Ectopic activation of *TMT3B* rescued wheat growth and yield penalties caused by MLO disruption[177]. |
| III | HOG02598 | *GW8* | *LOC_Os08g41940* | *GW8* modulates cell proliferation in rice[108]. |
| III | HOG03574 | *PROG1* | *LOC_Os07g05900* | *PROG1* encodes the zinc-finger nuclear transcriptional factor controlling prostrate growth in rice[178]. |
| III | HOG08247 | *SBEIIa* | *LOC_Os04g33460* | *SBEIIa* regulates sugary endosperm in rice[179]. |
| IV | HOG01435 | *KRN2* | *Zm00001d002641, LOC_Os04g48010* | *ZmKRN2* and *OsKRN2* encoding WD40 have regulatory roles in grain number in maize and rice, respectively[180]. |
| IV | HOG01513 | *RAVL1* | *Zm00001d002562* | *ZmRAVL1* encoding a B3-domian transcriptional factor can modulate the gene expression of brassinosteroid C-6 oxidase1[89]. |
| IV | HOG08376 | *DREB1C* | *LOC_Os06g03670* | *OsDREB1C* can regulate both photosynthesis and nitrogen utilization[181]. |

The types of I through IV are the same as those in Fig. 7c. Additional genes in orthogroups can be found in Supplementary Data 5.

representing subfamily-specific (or lineage-specific) retention (Fig. 8a; see representative genes in Table 2).

Oryzoideae are adapted to various aquatic environments, where roots encounter osmotic and other abiotic stresses. One possible response to osmotic stress is the deposition of wax to regulate root transcellular transport[79]. The synthesis of wax depends on fatty acid biosynthetic genes, such as *ACOT* (Acyl-CoA thioesterase), which belongs to a PROSOL-Or orthogroup, with Oryzoideae members retaining both rho-derived paralogues but the other three subfamilies having only one copy (Fig. 8b and Supplementary Fig. 54a; see the *ACOT* gene tree in Supplementary Fig. 54b). In addition, a role of *ACOT* in response to submergence stress is supported by the increased root expression of one of the rice *ACOT* paralogues after submergence[71] (Supplementary Fig. 54c), and by increased root expression under salt stress of *ACOT* homologues of the daisy-relative *Dendranthema grandiflorum*[80]. Pooideae include many members capable of diverse adaptability and important crops, such as wheat. One of the PROSOL-Po orthogroups is the casein kinase II (*CK2*) genes encoding proteins that included the regulatory subunit beta domain with diverse physiological roles in plants, such as light-signal transduction pathway and defense SA-mediated pathway[81]. Syntenic and phylogenetic analyses show the Pooideae *CK2β* genes retained two copies from rho (Fig. 8c and Supplementary Fig. 55a, b). Public RNA-seq data[82] showed different expression patterns of the wheat *CK2β* paralogues in response to cold stress (Fig. 8c and Supplementary Fig. 55c), implying that the *CK2β* genes might contribute to the adaptation of Pooideae species to various environments. Bamboos are characterized by fast-growing early shoots related to the anisotropic cell expansion of the rhizome lateral bud meristem[83]. Among the PROSOL-Bas, two are grass homologues of the *Arabidopsis SPIRAL1* (*SPR1*) gene (Supplementary Fig. 56a), which is required for the control of anisotropic cell expansion[84]. Phylogenetic analysis of the *SPR1* orthogroup (Supplementary Fig. 56b) further supported the hypothesis that Bambusoideae representatives have retained both rho-derived *SPR1* genes, but members of the other three subfamilies have retained only one copy (Fig. 8d). Furthermore, public RNA-seq data of rapidly-growing *P. edulis* bamboo shoots[85] showed that the expression of one *SPR1* homolog (*PHO2Gene20370*) was increased during shoot development,

whereas that of the other (*PHO2Gene45021*) was reduced, suggesting that the two *SPR1* paralogues have diverged in the regulation of in shoot growth (Fig. 8d and Supplementary Fig. 56c). For Panicoideae, *phosphoribosylanthranilate isomerase 1* (*PAI1*) genes, encoding a key enzyme in the tryptophan biosynthetic pathway, represent a PROSOL-Pa orthogroup with Panicoideae having retained both rho-derived paralogs (Fig. 8e, Supplementary Fig. 57a, b, and Table 2). Public RNA-seq data[86] showed different expression patterns of the sorghum *PAI1* paralogues in leaves under preflowering and postflowering drought stresses (Fig. 8e and Supplementary Fig. 57c). In maize, the *PAI1* gene was upregulated in the top and bottom crowns, which included meristematic cells for shoot and root tissues, under different cold stresses[87] (Supplementary Fig. 57d). Hence, the *PAI1* gene might contribute to plant acclimation under drought/cold stresses in Panicoideae.

In addition, 1296 orthogroups were retained as pairs in all four subfamilies, whereas 21 to 337 orthogroups had detected duplicates in two or three subfamilies (Fig. 8a). The orthogroups annotated for regulation of gene expression include several genes that were functionally analyzed in rice or maize with impact on grain yield, such as the rice genes *GROWTH-REGULATING FACTOR 4* (*GRF4*)[88] (Table 2), and the maize gene *RAVL1*[89] (Table 1). Furthermore, orthogroups with both copies retained include genes for environmental responses. For example, *MADS26* with both rho-derived duplicates (Table 2) is a negative regulator in rice in response to abiotic stress[90]. The retention of both rho-derived duplicates in all four subfamilies (Table 2) suggests their enhanced functions might have promoted grass evolution generally.

## Discussion

Our phylogenomic/phylotranscriptomic analyses here, including datasets from 363 grasses, provide support for rho and 17 other WGD events, of which five WGDs (#7–11) and 11 WGDs (#12–22) are placed in the large BOP and PACMAD clades, respectively, suggesting possible contribution of WGDs to the evolution of these highly diverse clades. In particular, the WGD event (#9; Supplementary Fig. 8) shared by *Deschampsia cespitosa* and *D. littoralis* is in agreement with a recent Pooideae study placing 3016 GDs at the MRCA of three *Deschampsia*

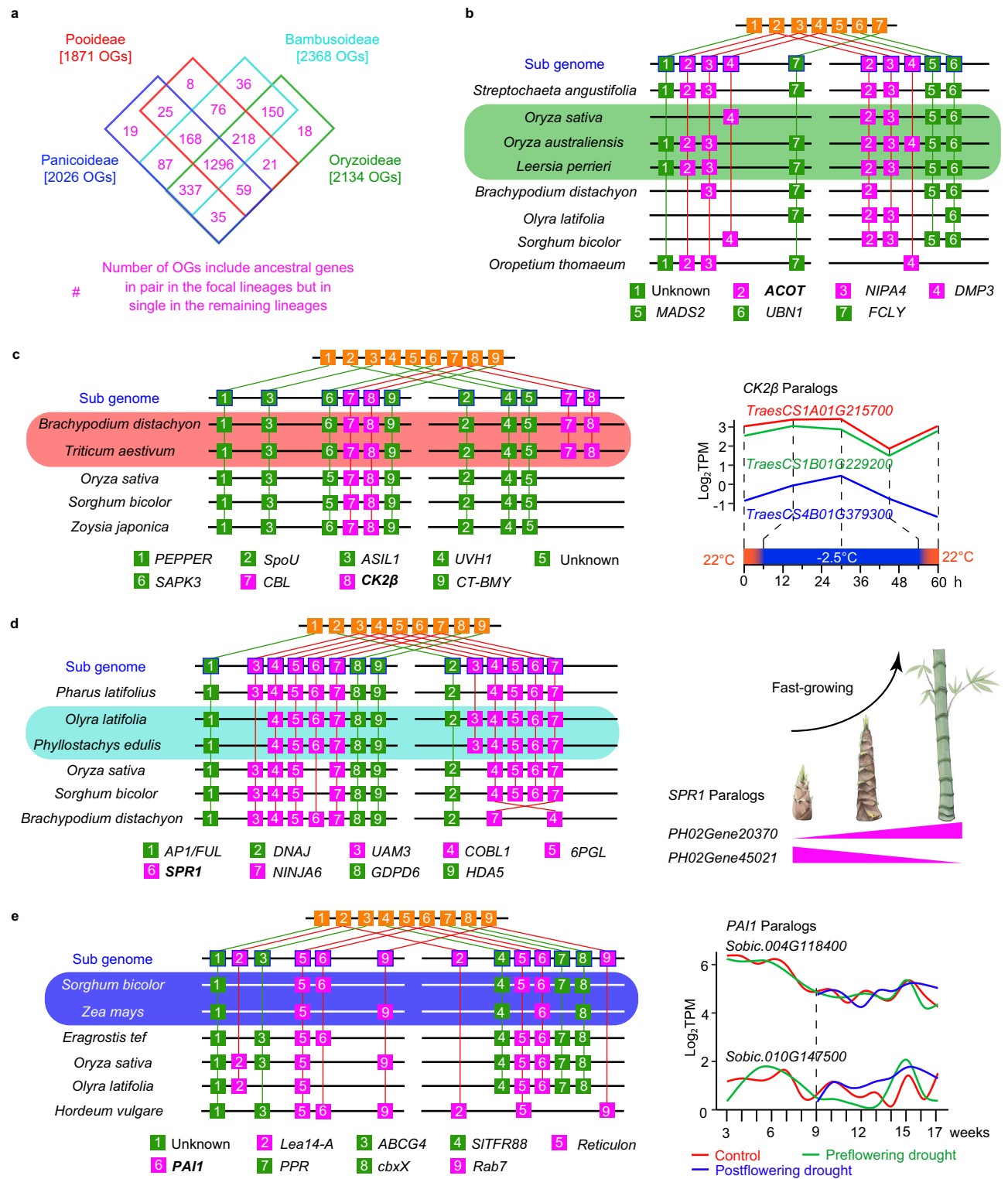

species[20] and consistent with the observed tetraploid cytotypes in several *Deschampsia* species[46]. In addition, GD detection using genomes and transcriptomes can increase the number of GDs at early branches in species-tree, because transcriptomes from the early diverging species can help to map gene pairs to more ancient positions. For example, among 1633 GDs mapped at Poaceae using 15 genomes, 9 transcriptomes, and 2 genome-skimming datasets (Supplementary Fig. 26a), 1010 and 728 GDs were shared by genes from the Anomochlooideae species *Streptochaeta angustifolia* (genome sequenced) and *S. spicata* (transcriptome sequenced), respectively.

Genome-skimming sequenced datasets of the Puelioideae species (*Puelia ciliata* and *Guaduella oblongifolia*) provide genes that shared 19–36 GDs of those 1633 GDs. Genomes tend to contribute to more duplicates (423–1018 GDs) than transcriptomes (313–728 GDs) and genome-skimming datasets with incomplete sequence and annotation. Sequenced genomes also allow comparison of gene orders of paralogs on chromosomes and hence provide GD evidence for WGD and SSD events. Integration of genomes and transcriptomes from basal lineages to core branches can provide GD clues for understanding gene and genome evolution. Furthermore, we estimated the

**Fig. 8 | Pair retained in one lineage but single-copy in other lineages (PROSOLs) and functional implications. a** Venn diagram showing the number of PROSOLs shared between or unique to Panicoideae, Pooideae, Bambusoideae, and Oryzoideae. **b** An illustration of gene retention in pair in Oryzoideae but as single-copy in representatives of three other subfamilies. The meaning of square and line colors in a syntenic block is the same as that in Fig. 7a. Gene annotations are indicated below the syntenic block. See details in Supplementary Fig. 54. **c** An illustration of gene retention in pair in Pooideae but as single-copy in representatives of three other subfamilies. The meaning of square and line colors in a syntenic block is the same as that in Fig. 7a. Gene annotations are listed below the synteny block. The line chart on the right of the syntenic block illustrates the expression patterns of the wheat *CK2β* paralogs in stems collected in five-time points and different temperatures. Red, green, and blue lines represent *TraesCS1A01G215700*, *TraesCS1B01G229200*, and *TraesCS4B01G379300*, respectively. See details in Supplementary Fig. 55. **d** An

illustration of gene retention in pair in Bambusoideae but as single-copy in representatives of three other subfamilies. The meaning of square and line colors in a syntenic block is the same as that in Fig. 7a. Gene annotations are listed below the synteny block. The expression patterns of two copies of *SRR1* genes in *Phyllostachys edulis* are illustrated on the right. See details in Supplementary Fig. 56. **e** An illustration of gene retention in pair in Panicoideae but as single-copy in representatives of three other subfamilies. The meaning of square and line colors in a syntenic block is the same as that in Fig. 7a. Gene annotations are listed below the synteny block. The graph on the right of the synteny block illustrates the expression patterns of the sorghum *PAI1* paralogues in leaves in 15 weeks in response to drought stress. Red, green, and blue curves represent control, preflowering drought, and postflowering drought, respectively. The dotted line indicates the flowering stage. See details in Supplementary Fig. 57. Source data are provided in a Source Data file.

divergence time of WGD events and placed 12 of 22 events near the Paleogene-Neogene boundary when dramatic climate changes occurred (#7, 10–16, and 19–22; Fig. 1 and Supplementary Data 3), supporting the idea that the WGDs might have helped grasses to survive under severe conditions, as proposed previously[1]. Similarly, the rho event was also mapped in a geological period with dramatic climate changes (#3; Fig. 1), suggesting a role of rho in adaptive evolution in grasses.

Our combined analyses of phylotranscriptomics and chromosomal positions uncovered WGD-related gene conversion and SSDs. Our phylogenomic/phylotranscriptomic analyses detected GD clusters at successive nodes on the species phylogeny, as also observed in other WGD analyses of different taxonomic groups, including genera, subtribes, tribes, subfamilies, families, orders, and at broader scales, such as angiosperms and other seed plants (e.g., refs. 1,2,36,48,91). We showed here, using syntenic analyses, that GDs at two or more nodes can correspond to gene duplicates in the same syntenic block, suggesting that they were generated by the same WGD event at an early node (Fig. 2c and Supplementary Fig. 30). The mapping of some GDs to a later node could be due to gene conversion, which can lead to equalization of different gene copies[49]. Gene conversions occur frequently in meiosis and has been hypothesized to play a key role in promoting polyploidy-dependent establishment of mutational robustness in plants[92]. In yeast, gene conversions were estimated to affect ~1% of the genome of each meiotic product per meiosis[93]. In plants, the estimates were $1.1 \times 10^{-5}$ per site per meiosis in *Arabidopsis*[94] and ~$3.3 \times 10^{-4}$ per marker per meiosis in rice[95]. Our phylogenomic analyses of bamboos uncovered 825 *Phyllostachys edulis* gene pairs that were likely derived from kappa (at the MRCA of woody bamboos) but mapped to the MRCA of temperate woody bamboos, suggesting they might have experienced gene conversion. The possible high gene conversion rate in woody bamboos might be related to their long reproductive cycle ranging from 30 to 60 years[96]. Moreover, we found that some GD clusters in Oryzoideae contain a large fraction of tandem duplicates (a form of SSDs) when chromosome positional information is available. This is similar to the recently reported GD cluster at the core Pooideae and other nodes[20]. Therefore, proposed WGD events supported by GD clusters from phylogenomic/phylotranscriptomic analyses should be further tested using assembled genomes when possible, with evidence from synteny and other chromosomal position information.

Analyses here provide genomic and evolutionary insights into progenitors of WBs and tetraploid rice. It has been proposed that WGDs often result from ancient hybridizations[21,97]; however, the parental lineages of many well-characterized WGDs are unknown, including rho. Here, we provide evidence that the WGD (kappa) shared by woody bamboos might have resulted from a hybridization involving a parental lineage related to herbaceous bamboos, but the other diploid parent might be extinct. Specifically, the number of GDs (1252) detected in the phylogenomic analysis [Analysis-I; with the topology of

(WB, HB) WB] is unusually large, considering that kappa was estimated to have occurred ~62 Ma. It is possible that the long generation time of woody bamboos has reduced the rate of gene losses that are often associated with most detected WGD events, which are largely supported by genomes of herbaceous plants. Hybridization among bamboos was previously proposed when an analysis using five plastid genes placed Olyreae and Bambuseae as sister (57% BS)[98], consistent with a recent study[99]; alternatively, hybridization was proposed to explain differences in plastid and nuclear phylogenies, with herbaceous bamboos as sister to tropical woody bamboos[53]. However, the monophyly of WBs was highly supported in a Poaceae phylogeny using >1000 nuclear genes[13], consistent with the hypothesis here that herbaceous bamboos are related to a progenitor lineage of all WBs.

Furthermore, our analyses support hybridization in the *Oryza* genus for the origin of the tetraploid *Oryza coarctata*, with one parent related to *O. brachyantha* (F genome) and the other being similar to the MRCA of the ABC and E genomes. Additional evidence for hybridizations in grasses were reported from analyses of gene phylogenies in several groups; for instance, the phylogeny of *pvcel1* homologues from tetraploid temperate WBs supports the hypothesis of hybridization of *Sasa* and *Phyllostachys* being the origin of *Hibanobambusa tranquillans*[53]. Our analyses revealed that 4959 GDs mapped at the MRCA of tetraploid *E. tef*[26] (2n = 4x = 40) and *Catalepis gracilis* in the Eragrostideae tribe (Chloridoideae) (#17; Supplementary Fig. 16) include 4821 GDs with the topology of ((*E. tef, C. gracilis*), *E. tef*), suggesting that *C. gracilis* might be related to a parental lineage of *E. tef*. Two previously proposed progenitors of *E. tef* were *E. pilosa* and *E. heteromera*[100], but their relationships with *C. gracilis* are unclear. Our analyses also show that 1164 of the 1779 GDs supporting a *Zoysia*-specific WGD (#18; Zoysieae, Chloridoideae) (Supplementary Fig. 14) has the topology of (*Z. japonica, Z. matrella + Z. pacifica*) (*Z. japonica, Z. matrella + Z. pacifica*), inconsistent with the previous hypothesis that *Z. matrella* was a hybrid between *Z. japonica* and *Z. pacifica*[42,101]. Our analyses placed 1481 GDs mapped at the MRCA of *Triticum* and *Aegilops*, representing a possible hybridization between them (Supplementary Fig. 10). Recent phylotranscriptomic analyses also supported a proposed scenario for the evolution of *Aegilops/Triticum*, including a possible hybridization between the A lineage (*T. urartu* and *T. boeoticum*) and the B lineage (*Ae. mutica*) that resulted in the D lineage (*Ae. caudata, Ae. umbellulata, Ae. tauschii*, and *Sitopsis*)[102]. Another study with phylogenetic analyses of four nuclear genes in Andropogoneae, a large tribe of Panicoideae, detected support for 28 tetraploidy events and 6 hexaploidy events among related species[21]. Therefore, allopolyploidization seems to have occurred repeatedly in grasses, with the possible hybridization for WB being a very ancient one.

Information on possible progenitors of polyploids can facilitate analyses of genome and gene function evolution, such as investigation of the functional impact of one subgenome (dominant) from differential gene retention and expression, in part, due to epigenetic regulation[103]. Our analyses of WB *BRD1* homeologues suggest that the

**Table 2 | Representatives of the PROSOLs genes**

| Subfamily | Orthogroup | Gene | Representatives | Function |
|---|---|---|---|---|
| Oryzoideae | HOGO2739 | ACOT | LOC_Os01g65950 | Expression of ACOT homologs increased in root under salt stress[80]. |
| Oryzoideae | HOGO6792 | bZIP79 | LOC_Os11g05480 | OsbZIP79 negatively regulates the phytoalexins metabolism[182]. |
| Oryzoideae | HOGO7296 | VQ10/31 | LOC_Os03g20330, LOC_Os07g48800 | OsVQ10 and OsVQ31 upregulated under the bacterial pathogen infection[183]. |
| Pooideae | HOGO4103 | MT-II-1a | TraesCS1D01G206300 | TaMT-II-1a is highly expressed in gain at the wheat ripening stage[107]. |
| Pooideae | HOGO4230 | CK2 | TraesCS1A01G215700 | TaCK2 was upregulated under drought stress in wheat[184]. |
| Panicoideae | HOGO2532 | PAI1 | Zm00001d053374 | PAI1 is highly expressed in rapidly diverging cells in Arabidopsis[185]. |
| Panicoideae | HOGO3905 | CKX3 | Zm00001d032664 | ZmCKX3 is highly expressed in the leaves of maize[186]. |
| Bambusoideae | HOGO0131 | SPR1 | PH02Gene20855, PH02Gene45021 | SPR1 gene is required for the control of anisotropic cell expansion in Arabidopsis[84]. |
| Bambusoideae | HOGO4648 | DWARF18 | PH02Gene43670, PH02Gene09039 | DWARF18 functions in GA biosynthesis to induce cell elongation in rice[187]. |
| Core Poaceae | HOGO0874 | IDD12/13 | LOC_Os08g36390 | OsIDD12 and OsIDD13 interact with SHR to modulate vein differentiation[188]. |
| Core Poaceae | HOGO1753 | GRF4 | LOC_Os02g47280 | GRF4, a target gene of OsmiR396, can regulate OsMYB61 that functions in cellulose synthesis in rice[88]. |
| Core Poaceae | HOGO2699 | MADS26 | LOC_Os08g02070 | OsMADS26 is a negative regulator of response to abiotic stress[90]. |
| Core Poaceae | HOGO5778 | NAC69 | TraesCS5D01G148800, TraesCS5A01G143200, TraesCS5B01G142100 | TaNAC69 can bind to promoters of the stress genes for modulating dehydration tolerance[189]. |

Additional genes in orthogroups can be found in Supplementary Data 5. The core Poaceae represent the four subfamilies retaining two copies.

homeologue dissimilar to that from HBs likely contributed to the rapid growth of WBs, suggesting that the unidentified parental lineage of the WB allopolyploid might have had rapid growth. *BRD1* encodes an enzyme catalyzing the final steps of brassinosteroid synthesis, and brassinosteroid was implicated in promoting bamboo shoot internode elongation in *Phyllostachys edulis*[104]. Furthermore, analyses of gene expression in *O. coarctata* revealed that some genes exhibited an increased expression due to the doubling of gene copies from both progenitors under full submergence in both purified water and salt water and that differential expression was observed for homeologues from different progenitors. Similar to previous studies of polyploids such as wheat, oat, and other allopolyploid crops (e.g., refs. 44,105–107), the results on allopolyploid grasses here provide an important resource for further (sub)genome analyses.

Rho-derived duplicates with different retention and loss patterns in subfamilies suggest functional evolution. The extensive analyses of retention and loss patterns of the rho-derived duplicates represent the first analysis for a large family using dozens of sequenced genomes and a cohesive nuclear phylogeny with >350 species, summarizing four different types (Fig. 7). Orthogroups with different retention/loss types were annotated with 48 GO terms of molecular functions. Among these GO terms, 13 terms were commonly detected in all four types, implying that different lineages have probably retained rho-derived duplicates in different orthogroups with the same or similar function(s). Additionally, GO analyses of 4,831 rice syntenic duplicates derived from rho revealed that 13 GO terms of molecular functions were significantly enriched, such as transcriptional regulation, hydrolase activity and other catalytic activity, protein binding, DNA binding, and ligand binding[16]. Dosage balance, neofunctionalization or subfunctionalization have been invoked to explain the retention of duplicates. Our analyses in Poaceae indicate that the rho-derived duplicates retained in the Type-III in long-term evolution (>133 million years) include genes related to development (such as *GW8* modulating cell proliferation in rice[108]) and that the rho-derived duplicates retained in the Type-I with lineage-specific duplications in short term evolution include genes related to stress response and adaption (such as *COLD1* for cold tolerance in rice[77]). Similar results were also reported in other plants, for instance, the *FHY3/FAR1* gene family with a role in tolerance to drought stress was expanded (via a recent WGD) in *Medicago ruthenica* genome[109]. In pomegranate, *CYP75* paralogues derived from an ancient WGD show different expression patterns in fruit development[110]. Furthermore, the detected PROSOL-Or, PROSOL-Po, PROSOL-Ba, and PROSOL-Pa genes provide a phylogenomic insight into subfamilial specific retention/loss patterns with possible contribution to diverse adaptations and morphologies, showing special value for understanding the retention/loss patterns of WGD-derived duplicates in plants.

In summary, the analyses here of patterns of gene duplication and losses from Poaceae WGDs highlight three main results: (1) support for lineage-specific WGDs, including kappa shared by woody bamboos; (2) phylogenomic evidence for gene duplicates from rho and kappa that experienced gene conversion, resulting in lineage-specific gene sequence evolution; and (3) lineage-dependent retention and losses of rho-derived duplicates, with implication for gene functional diversification and species changes in morphology and physiology. These results provide insights into the genome and gene function in Poaceae. As WGDs are associated with many large families and other large groups of angiosperms, these mechanisms for lineage-specific and WGD-related gene evolution might be general through angiosperm history.

## Methods

### Data source

To elucidate species relationships for more Poaceae representatives, we retrieved 319 grass datasets (315 transcriptomes and 4 genome-

skimming) that were reported in our previous studies[13,20] and integrated them with 36 other published grass datasets (29 genomes and 7 transcriptomes), representing 355 species (Supplementary Data 1). A combination of these 355 species, 16 datasets (15 transcriptomes and 1 genome) for other Poales, and nine genomes for other orders represented 380 species that were used for species-tree inference (see data accessions and references in Supplementary Data 1). These 380 species (including nine recently grass-published genomes other than transcriptomes) were used for molecular dating (see data accessions and references in Supplementary Data 1).

For WGD detecting in grasses via phylogenomic analyses, the same 355 grass species as above (with replacement of nine transcriptomes by recently published genomes) and eight additional recently published grass genomes were used; these datasets included 313 (= 315 + 7−9) transcriptomes, four genome-skimming datasets, and 46 (= 29 + 9 + 8) genomes and represented a total of 363 species, covering 45 tribes and 12 subfamilies of Poaceae (Supplementary Data 1). For WGD detecting in other Poales species via phylogenomic analyses, we used the 16 datasets for other Poales, the nine genomes for other orders, and two additional genomes [*Carex littledalei* (Cyperaceae; Poales)[111] and *Acorus tatarinowii* (Acorales; basal monocots)[112]]. For estimating the retention/loss patterns of the rho-derived duplicates among grass subfamilies, we also included the *Raddia distichophylla* genome[113] and the *Puccinellia tenuiflora* genome[114] (Supplementary Data 1). In addition, three other recently published *Aegilops* genome sequences[102] were used for the phylogenomic analyses in Triticodae (Supplementary Data 1).

Furthermore, to explore the gene expression patterns of interested genes using public data, we retrieved 15 wheat stem RNA-Seq datasets under cold stress[82] from NCBI (SRR22346048 through SRR22346062; Supplementary Data 1). The RNA-Seq datasets from sorghum leaves under control and drought during 15 weeks[86] were retrieved from NCBI (SRR8742861 through SRR8742957; Supplementary Data 1). We retrieved GSE167881 in GEO series of NCBI (Supplementary Data 1) to compare the maize gene expression under chilling[87]. We also used public bamboo RNA-seq datasets[85,115–117] from shoots, lateral bud, rhizome tip, leaf, and inflorescence (Supplementary Data 1). Furthermore, transcriptional and post-transcriptional change of rice *ACOT* (*LOC_Os01g65950*) gene under submergence over control was retrieved from the published data[71].

### Transcript assembly

Transcripts were de novo assembled and subsequently processed to remove the isoform (splice variants) and spurious sequences across transcripts. Briefly, RNA-Seq reads were processed by using Trimmomatic v0.32[118] to remove the reads with low quality and then assembled into de novo transcripts by using Trinity v2.2.0[119] with default setting. To remove putative contaminations from animals, humans, and bacteria in sampling, transcripts were blasted against the SILVA database (release_138.1 SSU)[120] by using BLAST v2.10.0[121] with the E-value threshold of $10^{-9}$, alignment length of ≥300 bp, and identity of >80%. Clean transcripts were annotated by using TransDecoder v5.5.0[122] to predict the coding regions. Finally, the longest open reading frame (ORF) among the ORFs in each transcript was extracted.

Contigs were de novo assembled by using SOAPdenovo v2.04-r240 with genome sequencing reads, and the contigs with length of <300 bp were removed. The retained contigs were searched against the reference genomes using Diamond v2.0.4.142[123] to identify the best hit that was used to predict ORFs using GeneWise v2-4-1[124]. The species with reference genomes are *Ananas comosus*[125] and nine grasses (including *Oryza sativa*[126], *Triticum aestivum*[107], *Zea mays*[127], *Phyllostachys edulis*[55], *Sorghum bicolor*[128], *Saccharum spontaneum*[129], *Hordeum vulgare*[130], *Thinopyrum elongatum*[131], and *Setaria italica*[132]).

To assemble fragmented transcripts into integral sequences and eliminate potential alternative splicing isoforms, chimeric sequences, and other redundant fragments, we employed previously described procedures[1,91] with minor changes. Briefly, the sequences of each species were clustered into groups according to their sequence similarity using a Markov clustering approach implemented in TransMCL v1[133], facilitating subsequent assembly of full-length transcripts based on a net-flow strategy. In this analysis, genes from species with (nearly) completely sequenced genomes were utilized as benchmarks to direct the assembly of genes derived from transcriptomic data. Subsequently, an SVM classifier in IsoSVM v2004[134], an in-built tool of TransMCL v1, was applied to discriminate between paralogs and isoforms based on overall sequence similarity and the distributions of insertions and deletions and single nucleotide polymorphisms. The sequences yielded from the above process were considered representative full-length transcripts for subsequent homologous comparison. In addition, to reduce noise in the estimation of WGD events shared by multiple species, species-specific transcripts were removed from each species following the BLASTp search against 40 monocot genomes and eight other angiosperm genomes (including *Arabidopsis thaliana*[135], *Malus × domestica*[136], *Punica granatum*[6], *Vitis vinifera*[137], *Solanum lycopersicum*[138], *Nelumbo nucifera*[139], *Nymphaea colorata*[140], and *Amborella trichopoda*[141]) (Supplementary Data 1). To reduce the computation time, we only selected the transcripts with a length of ≥300 bp for downstream analyses. Finally, to estimate the quality of final transcripts, we used BUSCO v5.2.2[142] and Monocotyledons-specific BUSCO database (liliopsida_odb10) to quantify completeness based on evolutionarily-informed expectations of the gene content of near-universal single-copy orthologs.

### Bayesian divergence time estimation

To estimate the divergence time across grasses, we used IQ-TREE v2.1.2[143] and 180 low-copy nuclear genes[13] to construct species-tree (Supplementary Fig. 1) and performed dating analyses with the MCMCTree tool in PAML v4.9[144]. To save computation time, we followed the method of summary of single gene family dating[145]. To meet the constraints of fossil records or broadly accepted dating results, we selected 166 gene sets to contain *Amborella trichopoda* for crown angiosperms, *Zostera marina* for crown monocots, *Ananas comosus* for crown Poales, and *Streptochaeta* for crown Poaceae. According to the recent report about angiosperm divergence time dating[146], we fixed the root of each gene tree to 209 MA. Other constraint fossil points can be found in Supplementary Fig. 2a. Then substitution rate was assessed by the baseml tool in PAML v4.9 with the GTR model. The MCMCTree analyses discard the first 50,000 iterations as burn-in, and then run the MCMC for 50 × 10,000 iterations, sampling every 50 iterations. The median value of ages was calculated under 95% confidence intervals. We used the deeptime package v1.0.1 in R to add geological timescales to the dated tree. The time-tree was compared with the paleo-climate changes during stratigraphic periods. The free-ice temperature was estimated by using the oxygen isotope δ18O content in fossils, which has been applied as a method to reflect the climate changes on geologic timescales[147]. The δ18O data were retrieved from previous analyses[147,148]. The dated tree and other phylogenetic trees in our study were visualized by using ggtree v1.14.6[149] in R.

### WGD identification through Tree2GD analyses

To identify WGD events shared by two or more species and place the published WGDs on the species-tree, we used Tree2GD v1.0.40[4] to place GDs on the species-tree. To reduce computation time, we assigned subsets of species to different phylogenetic groups and focused on the WGD events shared by multiple taxa in a specific group (see groups in Supplementary Figs. 3–16). For each group, we used diamond v2.0.4.142[123] to perform all-by-all BLASTp (parameters: --more-sensitive --max-target-seqs 20 --evalue 1e-5 --masking 0). The resulting gene families were processed by a Markov cluster using

PhyloMCL v2.0[150] with default parameter values to identify orthologous gene families, of which orthologs with ≥5 genes from ≥2 species (≥2 genes from one ingroup species) were selected for further analyses to reduce computation time. In addition, each orthologous group was aligned by PASTA v1.8.5[151] (three iterations, each with (1) tree search using FastTree v2, (2) sequence alignment for each clade in the tree using MAFFT v7.372[152], and (3) merging of all alignments using MUSCLE v3.8.425[153]). The produced protein alignment were back-translated into nucleotide sequences by using the PAL2NAL script (v14) in Perl[154]. Each nucleotide alignment was trimmed by using trimAl v1.4.rev22 (parameters: -automated1 -resoverlap 0.7 -seqoverlap 75) to remove gaps and sequences with low coverage. The trimmed nucleotide alignments were used to reconstruct gene trees by using IQ-TREE v2.1.2[143] with the ML method, GTR model, and 1000 ultrafast bootstrap replicates[155]. Finally, we used Tree2GD (parameters: --species=2 --bp=50 --root=MAX_MIX) to reconcile gene trees with species-tree and estimated the number of GD events as defined previously[4].

Quantitative detection of GD clusters from phylogenomic analyses is commonly used to support a candidate WGD on a specific branch of species-tree, with the number of GDs observed higher than the expected number[1,3,4,47,48,91]. WGD-derived paralogs sometimes can be largely retained in both subclades and form a (AB)(AB) gene topology. Thus GDs of the (AB)(AB) type provide signals for candidate WGDs shared by two or more species[4,91]. In addition, possible misinterpretation of GDs from dramatic variations of evolutionary rates among lineages (corresponding to long branches) in gene trees can be avoided by expansion of taxon sampling[47]. A recent phylogenomic study of 68 angiosperm genomes discovered hundreds of GDs in the (AB)(AB) type, consistent with the phylogenetic placement of each published WGD[4]. Following these phylogenomic analyses, here we focused on the GD clusters with >200 GDs (a cut-off based on the lowest number of GDs for well-established WGDs in genomes[4]) and proposed these clusters with relatively high numbers of GDs in the (AB)(AB) type to be candidate WGDs for further estimation.

## WGD identification through MAPS analyses

To measure the statistically significant difference between the ratio of retained GDs from ancient WGDs or SSDs that occurred over a much longer window of time, we analyzed the rho, sigma, and tau events, according to the minimum number of required species in the ladder species-tree for MAPS and the effective estimation for ancient WGDs as in previous cases using MAPS[1,2,48]. For a particular WGD, a tree with a nested subtree corresponding to this WGD were generated by deleting taxa from the dated species-tree (Supplementary Fig. 2). All formed ladder species trees were required to contain at least five species (including one outgroup; Supplementary Fig. 25). Gene families that include at least one gene copy from each taxon and at least two gene copies from at least two species were identified by PhyloMCL v2.0[150] using sequences from all-against-all BLASTp searches with diamond v2.0.4.142[123]. In addition, sequence alignment and tree construction for all gene families are the same as that in the section on "WGD identification through Tree2GD analyses". Gene trees were mapped onto their corresponding species-tree by using MAPS (parameters: --mt 40 --mb 50). The subtree duplication rate was compared to that inferred from a null simulation which assumes no WGD event. For the null simulation, gene birth (λ) and death (μ) rates were predicted by WGDgc v1.3[156] (mMax = 100) in R [the geomMean (φ), λ and μ values for each group are shown in Supplementary Fig. 25]. A total of 3000 simulated gene trees for each dated species-tree were generated by GenPhyloData in JPrIME v0.3.7[157], including 1000 trees for half of gene birth and death rates, 1000 trees for three times of gene birth and death rates, and 1000 trees for observed gene birth and death rates. Then 1000 trees were sampled 100 times from these 3000 trees and mapped onto their respective species-tree to survey subtree duplication rate.

## WGD inference using divergence time of paralogous gene duplications

Peaks of the Ks distribution of paralogues (Ks-plot) have been accepted as signals of GD bursts and applied to WGD identification[1,2]. To infer candidate WGDs in each taxon with transcriptome or low-coverage genome datasets, we utilized a method that has been included in Tree2GD v1.0.40[4] and is similar to the Node-Ks approach used in previous studies[1,2]. Briefly, we selected the paralogues shared a GD (BS ≥50) from the Tree2GD results and used MUSCLE v3.8.425[153] to align the protein sequences of each paralogue pair. The protein alignment was back-translated into nucleotide sequences by the PAL2NAL script v14[154]. The resulting nucleotide sequences without gap were used to calculate the Ks value through the codeml in PAML v4.9[144] with a maximum likelihood method of GY[158]. The Ks values greater than five were not used in subsequent WGD analyses to avoid pitfalls of Ks saturation[1,2], and the median Ks value of all paralogues from a common GD event was used as the Ks value of the event to reduce the effect of multiple copies on Ks-plot. Each peak in Ks-plot as an event was identified, and the median value was calculated as the Ks value of the event[1].

Furthermore, to place a WGD event on the species-tree, we compared the divergence time of paralogues from a focal taxon, that of orthologues from the taxon and the other one shared the event, and that of orthologues from the taxon and one without the event[1]. Briefly, orthologue pairs were identified by searches using diamond v2.0.4.142[123] with the reciprocal best hit (RBH; "getRBH.pl" available in https://github.com/Computational-conSequences/SequenceTools/) between two taxa[159]. In addition, orthologues were aligned and back-translated into nucleotide sequences using the same method as described above. Finally, we used the codeml with GY method to calculate the Ks value of each orthologous gene pair and retrieved the median Ks values of orthologs. The orthologue Ks peak values were compared with the Ks value of the putative focal WGD shared by the ingroup to estimate whether the WGD occurred earlier than the speciation between lineages shared the WGD event and later than the divergence of their stem group from an outgroup (Supplementary Figs. 17–23).

## Chromosomal collinearity analyses

To identify WGD events in a taxon with genome sequenced, we applied the MCScan pipeline (Python version)[160] in JCVI v1.1.15 with a C-score cutoff of 0.5 to identify the chromosomally collinear blocks, which contain at least four collinear (syntenic) gene pairs. Ks values of each syntenic gene pair were calculated using the same methods in the above section, and the median Ks value of all gene pairs in a syntenic block were used as the Ks value of the block. To identify the type and age of WGD events during evolution, the different blocks with the dating in the same range clustered by a Ks peak were marked with the same color in genome dot-plot analyses. Furthermore, to place a published WGD event identified in a single genome-sequenced species into the grass phylogeny, we estimated the number and phylogenetic positions of GDs shared by the gene pairs anchored in syntenic blocks. Finally, when analyzing genes in inter-species collinear blocks, to reduce redundant orthologous signals in an orthologous group (OG) that resulted from tandem duplications in the outgroup, we defined $T_{OG}$ as a cumulative value of the product of percent identity and alignment coverage for sequences in an OG and calculated it using Eq. (1) as shown below:

$$T_{OG} = \sum_{i}^{n} \left( Identity_{Outsp-Insp_i} \times \frac{AL_{Outsp-Insp_i}}{GL_{Insp_i}} \right) \quad (1)$$

$$T_{max} = \max(T_{OG1},,,T_{OGm}) \quad (2)$$

where the percent identity ($Identity_{Outsp\text{-}Insp_i}$) between an outgroup gene (Outsp) and each ingroup sequence in an OG with $n$ ingroup genes ($Insp_i$) is multiplied by the ratio of alignment length (AL) for Outsp with each $Insp_i$ (Outsp-$Insp_i$) over the gene length (GL) of $Insp_i$ in an OG; $T_{max}$ represents the maximum value among $T_{OG}$ of $m$ tandemly duplicated genes. Then we selected the OG with $T_{max}$ (from Eq. 2) for downstream analyses.

## Estimation of the rho-derived duplicates in subfamilies

To estimate the retention and loss patterns of the rho-derived duplicates in different subfamilies, we integrated inter-species collinear blocks and gene trees to analyze the genes retained from rho. We selected 24 genomes that represent seven subfamilies and the pineapple genome as an outgroup (see the species in Supplementary Fig. 48 and Supplementary Data 1). These genomes were compared by MCScan to identify intra- and inter-species collinear blocks based on three relationships (see pipeline in Supplementary Fig. 48). (1) we required relationships between interspecific blocks with a match of one pineapple block versus two blocks per grass species that did not undergo a recent WGD after rho (including *Oropetium thomaeum*, *Cenchrus americanus*, *Setaria italica*, *Saccharum hybrid*, *Sorghum bicolor*, *Thinopyrum elongatum*, *Hordeum vulgare*, *Puccinellia tenuiflora*, *Brachypodium distachyon*, *Raddia distichophylla*, *Olyra latifolia*, *Oryza sativa*, *O. officinalis*, *O. brachyantha*, *Leersia perrieri*, *Pharus latifolius*, and *Streptochaeta angustifolia*). (2) we required relationships between interspecific blocks with a match of one pineapple block versus four blocks in grass species that have undergone a WGD after rho (*Zoysia japonica*, *Phyllostachys edulis*, *Zea mays*, *Eragrostis tef*, and *Zizania latifolia*). (3) we required relationships between interspecific blocks with a match of one pineapple block versus six grass blocks for each of *Dendrocalamus latiflorus* and *Triticum aestivum*. These interspecific blocks were integrated according to the gene orders in the pineapple genome. The integrated blocks were split into continuous blocks based on the chromosome-scale assemblies from pineapple, *P. latifolius*, *O. sativa*, *L. perrieri*, *B. distachyon*, *T. elongatum*, *H. vulgare*, *P. tenuiflora*, and *S. bicolor*. In addition, to ensure the block supporting the rho event, we required that the block include at least one pair of paralogous genes mapped at Poaceae and/or anchored in the syntenic blocks with some genes matching the rho event. Therefore, such gene pairs detected from the gene trees with more grass and outgroup representatives (see duplications in Supplementary Fig. 3) were applied as markers to filter blocks (see a sample of block in Fig. 7a and Supplementary Fig. 49). In the retained collinear blocks, the redundant orthologous signals resulted from tandem duplications in the pineapple genome were removed using the $T_{OG}$ and $T_{max}$ as described in the section of "Chromosomal collinearity analyses".

To estimate the retention and loss patterns of the rho-derived duplicates, we construct gene trees of the orthogroups that were defined here as the pineapple gene and its collinear grass genes in the retained collinear blocks. Specifically, protein sequences of each orthogroup were aligned by using the above PASTA approach. The produced protein sequence alignments were back-translated into nucleotide sequences by using the PAL2NAL script v14[154]. The nucleotide alignments were used to reconstruct gene trees by using IQ-TREE v2.1.2[143] with the ML method, GTR model, and 1000 ultrafast bootstrap replicates[155]. In addition, genes with possibly false or incomplete assembly and annotation could result in deletion or insertion regions in multiple sequence alignments and long terminal branches in gene trees. Uncorrected positions of the long terminal branches could bring false positive results of GD mapping. To reduce the noise signals from long branches, we iteratively pruned the long terminal branches that were 8 times longer than the average value (in 95% confidence interval) of the length of retained branches in gene tree until the average value did not change. The noise signals in multiple sequence alignments can also lead to a long interbranch that

connects terminal branches; hence we also removed the long interbranch that was 8 times longer than the average length of branches in the gene tree. The (pruned) gene trees (including at least 6 genes) were rooted by using the minimal ancestor deviation (mad v2.2[161]), a method based on branch lengths. When one or more duplications shared by at least three subfamilies were detected in the rooted gene tree, the gene tree were iteratively rooted with each node (and the pineapple gene) as an outgroup to keep the minimal ancestral duplication.

The final rooted gene trees were reconciled to species-tree to estimate the number of retention and loss events (including the species-specific reciprocal loss of two rho-derived copies that were retained in a subfamily). To detect possible retention after a duplication, we required that the duplication (BS ≥ 50) was shared by at least three gene pairs. When a Poales duplication was detected in a gene tree, the tree was pruned into two subtrees from the duplication, one with the pineapple and grass genes and the other including the grass genes. Similarly, when two or more duplications were mapped at Poaceae and the nodes (the MRCA of Pharoideae and core Poaceae, the MRCA of core Poaceae), with duplications mapped, had equal depth (or <3 depth differences), the gene tree was pruned into subtrees from the duplications. The subtrees including at least six genes from at least four subfamilies were respectively used to estimate the retention and loss events after rho in Poaceae. A Poaceae OG was defined as a clade of Poaceae genes after the divergence of non-Poaceae families of Poales; if a Poaceae OG has a GD mapped to one of the backbone nodes from the MRCA of Poaceae to the MRCA of core Poaceae, then this OG is defined as having two rho-derived copies. If a Poaceae OG lacks such a GD, then it is defined as being single-copy for rho duplicates. The number of retained genes in subfamilies are in Supplementary Data 5.

Furthermore, to explore the gene expression patterns of interested genes in the Poaceae OGs, we used kallisto v0.46.1[162] to quantify and compare the gene expression levels using public data (Supplementary Data 1).

## Bamboo genome analyses

We performed different analyses to investigate the Kappa event. Analysis-I was a phylogenomic analyses using multiple species, representing five Olyreae genera (one with sequenced genome plus four with transcriptomes), 14 genera of Arundinarieae (one with sequenced genome and 17 with transcriptomes), and 14 genera of Bambuseae (three with genomes and 23 with transcriptomes) (Supplementary Fig. 5). Using the above phylogenomic approach in Tree2GD analyses, gene trees of gene families were constructed and then reconciled with species-tree to detect GD events. For GDs mapped at woody bamboo ancestor, syntenic genes matched the GDs with different retention types were examined to investigate their presence in syntenic blocks (Supplementary Fig. 31). In addition, analysis-II used MCScan to identify collinear blocks with the relationships of one outgroup (*Oryza sativa* and/or *Thinopyrum elongatum*) versus one *Olyra latifolia* versus two *Phyllostachys edulis* versus three *Dendrocalamus latiflorus*. In each block, orthologous groups with tandem duplicates in single species were filtered out by using the $T_{OG}$ and $T_{max}$ as described previously. Orthologous groups were aligned by MAFFT v7.372[152] and gene trees were constructed by IQ-TREE v2.1.2[143]. Gene trees are rooted with non-bamboo grasses and reconciled to species-tree to map GDs shared by syntenic genes (Supplementary Fig. 32). We compared the syntenic gene pairs mapped at the MRCA of woody bamboo ancestor with the GD evidence here and that with the evidence from analysis-I (Supplementary Fig. 32). Moreover, using the above approach in Ks analyses, analysis-III dated the Ks peak from *Phyllostachys edulis* syntenic gene pairs for the Kappa event (Supplementary Fig. 20b and Supplementary Data 3). We also compared the number of syntenic gene pairs with Ks evidence for kappa and that with GD evidence for kappa from analysis-I (Supplementary Fig. 20b).

For GDs mapped at Bambusoideae, we detected their corresponding syntenic genes and examined their presence in the syntenic blocks that also include genes of GDs mapped at the MRCA of wood bamboos (Supplementary Fig. 33). In addition, if paralogues from woody bamboos have unusually high substitution rates (high evolution rate), such genes could be placed as sister to bamboos due to long-branch attraction (LBA) artifacts[56], resulting in incorrect placement of the GD at Bambusoideae. To examine the potential effects of LBA artifacts in the detection of GDs mapped at Bambusoideae of the (AB)A type, 242 gene trees with non-bamboo grasses as outgroup are reconstructed by using the first+second codons from the nucleotide alignments in Analysis-I. These gene trees were reconciled with species-tree to detect GDs (BS ≥50); these GDs were then compared to those obtained previously by the paralogues mapped at Bambusoideae using full codons. The number of GDs shared by two or more species were shown in phylogeny of Supplementary Fig. 34b. Among the gene trees with GDs mapped at Bambusoideae using the 1st+2nd codons and the gene trees of the same gene families using full codons, we used student's Fisher test to compare the significant difference between the branch length of herbaceous bamboo lineage (III) and each of other bamboo lineages [including two woody bamboo lineages (I, II) and the Bambusoideae lineage (IV)] (Supplementary Fig. 34).

To detect possible genome regions related to hybridization, using the above MCScan approach, the *O. latifolia* genome was aligned against the *P. edulis* or *D. latiflorus* genomes to identify inter-species collinear blocks between herbaceous bamboos (HB) and woody bamboos (WB) (Supplementary Fig. 35). For each of the inter-species collinear blocks, we compared the two homeologous chromosomes of WB to count the number of single-copy genes between HB and WB (SCG); according to the number of SCGs, the chromosomal fragments with more SCGs were named as the dominant subgenome (SCG H-D) and the chromosomal fragments with less SCGs were named as the recessive subgenome (SCG H-R). We compared each pair of homeologous chromosomes for the number of gene pairs of WB paralogs, SCG H-D, and SCG H-R and used "aov" in R to examine significant difference among them (Supplementary Fig. 36).

### *Oryza* genome analyses

To place *Oryza coarctata* in the *Oryza* phylogeny, we utilized ASTRAL-Pro v1.3.1.0 to infer phylogenetic relationships with 22,829 orthologous groups, which contain at least two copies in at least one species and are used to identify WGDs in Oryzoideae. To infer the probably parental subgenomes of *O. coarctata*, we estimated the number of sisterhood (lineage sister to *O. coarctata*) in gene trees. About 5657 gene trees that contained two main sisterhoods were pruned to remain single-copy for all species except for *O. coarctata* and to reduce the effect of gene duplications (especially reciprocal retention and loss of duplicates in different lineages). (1) a monophyletic Poaceae is required (if a GD was mapped at Poaceae, each subclade contained at least one focal sisterhood was retained). (2) only one gene for each outgroup species was saved by iteration of searching clades with duplicates and removing one of the duplicates with a longer branch length (and/or removing one of the duplicated groups with less species coverage). (3) only one gene for Oryzoideae species (except for *O. coarctata*) was saved by the same procedure in the second step [if one of the duplicated clades (sharing two or more species) containing *O. coarctata* genes excluded either of the two focal sisterhoods, this clade was removed]. The retained gene trees were reconstructed by using RAxML and then used to infer *Oryza* genome phylogeny by using ASTRAL. The RNA-Seq data[64] of *O. coarctata* under control (SRR771527), under salt-water submergence (SRR771531), and under purified-water submergence (SRR771530) (Supplementary Data 1) are used for calculating transcripts per million values by kallisto v0.46.1[162]. In addition, we retrieved the published expression data[71] of rice root under submergence.

### GO analyses

An orthogroup represents a set of homologous genes, with similar functions derived from their common ancestor, and a GO term of their consensus sequence indicates a possible function of the orthogroup. To compare GO annotations of orthogroups, a consensus sequence of each orthogroup was generated used the hmmemit tool in HMMER package v3.4[163] with protein sequence alignments of each orthogroup and the simple majority rule. In addition, we applied the online Inter-Pro program (https://www.ebi.ac.uk/interpro/search/sequence/) to predict GO terms using consensus sequences to search for homologous proteins in the default databases with proteins of several organisms and mapping the resulting gene hits to InterPro2GO database[164–166]. Among GO terms annotated by using currently active GO information (basic-go database version: releases/2023-11-15; http://purl.obolibrary.org/obo/go/go-basic.obo)[167,168], we selected the ones belonging to the 'Molecular Function' category for downstream analyses.

To obtain a term from multiple GO terms that were derived from an ontology and annotated on an orthogroup, we calculated the (semantic similarity) distance between any specific term and its ancestor(s) of the third depth and select the ancestral term with the shortest distance[169]. To simplify the GO classification, we grouped the GO terms into the regulation of gene expression, protein regulation, and modification, metabolism, small-molecule metabolism, nucleic acid metabolism, interaction of proteins, transport, and other categories. To implicate special functions of the rho-derived gene duplicates, we performed gene category analyses by comparison of the GO terms for the duplicates and the whole protein-coding genes in each of five grass genome (rice, barley, wheat, maize, and sorghum) with GOATOOLS v1.3.9[170] (Supplementary Data 6). The GO terms of the five genomes were downloaded from PLAZA v5.0[171]. Under Fisher's exact test and multiple test correction, the GO terms with $p$ values <0.05 were selected as enriched terms. In addition, we used clusterProfiler v3.10.1[172] for GO analyses of bamboo genes (Supplementary Fig. 38a).

### Statistical analyses

We applied different functions in R to perform statistical analyses, including "t. test" for the Student's $t$-test, "Fisher.test" for Fisher analysis, "cor.test" for Pearson's correlation analysis, "chisq.test" for the Pearson's chi-squared test, and "aov" for the Exact F-test.

### Reporting summary

Further information on research design is available in the Nature Portfolio Reporting Summary linked to this article.

## Data availability

The accessions in Supplementary Data 1 are available in NCBI and other public databases. Datasets including the sequence alignments for molecular dating, sequence alignments and gene tree files of orthogroups and their reconciliations for Tree2GD analyses, sequence alignments and gene tree files for MAPS analyses, sequence alignments and gene tree files for bamboo genome analyses, gene tree files for *Oryza* ASTRAL analyses, and sequence alignments and gene tree files for estimating the retention and loss patterns of the rho-derived duplicates, are available on FigShare[173]. The relevant data for Figs. 1–8 can be found in the Source Data file. Source Data for Supplementary Figs. are also provided in the Source Data file. Specific databases used in our analyses include the SILVA database (releases/24-Aug-2020; https://www.arb-silva.de/fileadmin/silva_databases/release_138_1/Exports/SILVA_138.1_SSURef_tax_silva_trunc.fasta.gz), the Monocotyledons-specific BUSCO database (liliopsida_odb10; https://busco-data.ezlab.org/v4/data/lineages/liliopsida_odb10.2020-09-10.tar.gz), and the basic-go database (releases/2023-11-15; http://purl.obolibrary.org/obo/go/go-basic.obo). Source data are provided with this paper.

## Code availability

The custom scripts for estimating the retention and loss patterns of the rho-derived duplicates in different subfamilies are available at https://github.com/TaikuiZhang/GrassPhylogenomics and at https://doi.org/10.24433/CO.1170454.v1.

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

## Acknowledgements

We thank Drs. Guojin Zhang, Jun Wang, Pengfei Ma, Yunlong Liu, and Cuiyu Liu for valuable discussions; Drs. Shengyu Zhou and Jingting Shen for providing the TransMCL software; and Yaning Shi for the plant drawings in Figs. 2 and 8. We thank Dr. Junpeng Shi and two anonymous reviewers for very helpful comments on the manuscript. Computations for this research were performed on the Pennsylvania State University's Institute for Computational and Data Sciences' Roar supercomputer (USA), the National Supercomputing Center in Shenzhen (China), and the Computing Center of Beijing Institutes of Life Science (China). This work was supported by funds from Eberly College of Science and the Huck Institutes of the Life Sciences at the Pennsylvania State University (H.M.), the National Natural Science Foundation of China (32070247 to J.Q.), and the China Postdoctoral Science Foundation (2019M661344 to T.Z.).

## Author contributions

H.M., J.Q., and T.Z. designed the research. H.M. and J.Q. supervised research. W.H. and L.Z. contributed grass transcriptome datasets. T.Z. performed analyses and wrote the draft manuscript. H.M., J.Q., D.-Z.L., L.Z., and W.H. commented on the results. H.M., D.-Z.L., and L.Z. revised the manuscript. H.M., J.Q., and T.Z. provided funds.

## Competing interests

The authors declare no competing interests.
