## [Peer Review File · Nature Communications]

Phylogenomic Profiles of Whole-Genome Duplications in Poaceae and Landscape of Differential Duplicate Retention and Losses among Major Poaceae LineagesReviewers' Comments:

Reviewer #1:

Remarks to the Author:

Zhang and his colleagues presented a very comprehensive profiling of the Whole-Genome Duplications (WGDs) events in Poaceae. They also reported the detailed WGD analysis in woody bamboo and *Oryza*, and finalized their work by investigating lineage specific differential retention of rho-derived duplicates and their potential functional consequences. Overall, this is a nice work that offers many novel evolutionary insights into Poaceae. One of my major concerns was that the manuscript was too long to effectively capture the key points. For example, the authors spend six pages, three sub-titles to report the major finding of the paleo-polyploidization (called kappa) event in woody bamboo, and most of the analysis in these sections were to test the topology of GDs in different lineages that supported this kappa polyploidization. I suggest the authors to re-organize these sections and simplify their writings to make this manuscript more readable. I also have some further comments below:

1. Line 224-229, the authors applied Ks analysis to date the time of WGDs. They used the example in *Ischaemum*, with the Ks of WGD paralogs, orthologs between *Ischaemum* species and orthologs between *Ischaemum* and *Eulaliopsis* being 0.1144, 0.0599 and 0.1184, respectively. In my opinion, these data suggested that the WGD in *Ischaemum* lineage happened soon after its split with *Eulaliopsis*, so I cannot understand the author's conclusion in Line 255 that the duplications in a burst was from a single event?

2. The authors used the gene sequences from both genome assemblies and transcriptome assemblies to perform this phylogenomic study. As the authors have stated (Line 661), the completeness of gene annotation may affect the examination of TDs. Similarly, the completeness of transcript assemblies due to the limited RNA-seq data (Table S2) may also create bias during the identification of GDs. It's a good opportunity to evaluate the strength and weakness of using transcript assemblies to identify GDs, or at least make some discussions. By the way, I also strongly suggest the authors to divide the Results and Discussions into two different sections, and put some discussion points that currently mixed with results into the new Discussions section.

3. Line 744, the authors carried out the analysis of rho-derived PROSOL genes, and reported the lineage specific PROSOL genes in Panicoideae, Pooideae, Bambusoideae, and Oryzoideae. These analyses were based on the genes that clustered into Orthogroups. Since lineage specific WGDs after rho were common in all these four subfamilies, my question was how to exclude the misidentification of rho-retained duplicates, which might actually come from the reciprocal loss of duplicate genes after lineage specific WGDs. For example, the FIE1 and FIE2 genes were reported to come from the reciprocal loss of one of their paralogs after maize WGD (Swigonova, et al., *Genome Res*, 2004.).

Minor:

1. Line 234, "WGD dating can be affected by the ingroup species that have evolved more quickly (with greater mutation rates) than outgroup species". Both the acceleration or decline of mutation rate may affect the estimation of molecular dating.

2. Line 155, a typo of "...GDs with gene duplicates from in sequenced Poaceae genomes"

3. Line 582, a typo of "...pruned the gene trees to by removing putative paralogs"

Reviewer #2:

Remarks to the Author:

Dear Editor I have revised the manuscript titled Phylogenomic Profiles of Whole-Genome Duplications in Poaceae and Landscape of Differential Duplicate Retention and Losses among Major Poaceae Lineages by Taikui Zhang, Weichen Huang, Lin Zhang, De-Zhu Li, Ji Qi, and Hong Ma.

Whole-genome duplications (WGDs) are important drivers of angiosperms evolution and diversification. Previous studies reported three WGDs in the ecologically and economically important grass (Poaceae) family; however, these studies included a limited number of species, suspecting that there are more WGDs that has not been discovered yet. In the present work, authors performed

phylogenomic/phylotranscriptomic analyses of 363 grasses covering all 12 Poaceae subfamilies to detected strong evidence for new WGDs, and other Gene Duplication clusters (GDs) and explore their possible roles in adaptive evolution and species divergence.

The paper is well written and I enjoyed the combination of methods to address, with confident, new events of WGDs and calibrate the timing of such events. From my perspective this is a very interested paper that significantly contribute with the knowledge of genome duplications presenting a whole picture of several duplication events in grasses. Sampling and methods are very well chosen and documented. Authors did an excellent selection of figures that support the main text. The manuscript presents a detailed literature compilation.

I am not a native speaker; however, in my opinion, some minor language issues need to be addressed. Specially, pay attention to large sentences.

Reviewer #3:

Remarks to the Author:

The manuscript presents a significant contribution to the field of evolutionary biology, specifically in the realm of phylogenomics and whole-genome duplications (WGDs) in Poaceae. The findings are of considerable value to the scientific community, shedding light on lineage-specific WGDs, gene sequence evolution, and the implications of lineage-dependent duplicate retention and losses. The use of syntenic analyses to correlate gene duplicates from the same WGD event is a commendable approach. However, there are notable areas that require attention, including the need for subheaders to help clarify the paper's outline. The manuscript holds great promise and significance but requires revisions to address the outlined concerns. The incorporation of suggested improvements will contribute to the overall quality and clarity of the research.

General comments:

1. Adding subheaders would enhance the manuscript's organization, providing better guidance for readers. Additionally, the outline of the paper is not entirely clear. A more structured presentation of the research would improve the overall readability. A proposed revision involves incorporating dedicated sections that accentuate some of the newly described WGD events which are discussed at length. These sections could encompass detailed methodologies employed to confirm their placements, coupled with an exploration of intriguing gene retention patterns observed during these events.
2. The manuscript makes excessive use of acronyms, which may hinder comprehension for readers. It is especially difficult to keep track of the HB, WB, TWB, NWB, and PWB when discussing the ABCD genomes.
3. The manuscript should explicitly address how the current work builds on the authors previous paper on phylogeny, especially concerning the section on polyploidy. Providing this connection will enhance the contextualization of the research within the existing literature.
4. I really appreciate the authors detailed explanation of the PROSOL genes, but I was wondering if this could possibly be condensed to a table with a shorter summary in the main text. Currently, the paper is over 30 pages, and this could be a way to shorten it.

Specific comments:

Line 15 – the first sentence of the abstract is hard to follow, please consider breaking into multiple sentences.

Line 43 – I don't think the authors mean molecular dating placed the WGD events, but rather some other analyses did?

Line 88 – “Additionally, rice and other *Oryza* species collectively have 11 reported genome types (six diploids and five allotetraploids); furthermore, domestic and wild *Oryza* species that have adapted to different aquatic environments.” – this sentence could be broken up for clarity.

Line 97 – Can the authors please clarify what they mean “successive species phylogenetic positions”

Line 129 – this paragraph ends with recently and the next paragraph starts with the same word.

Line 155 – Can the authors please clarify what they mean by “For those WGDs that are supported by GDs with gene duplicates from in sequenced Poaceae genomes, we further estimated the number of GDs with detected duplicates in syntenic blocks (collinear genomic regions).”

Line 216 – I am not sure that Ks analyses can be referred to as molecular dating. I may be incorrect, but they are all relative, not absolute, so perhaps avoiding calling them molecular dating would be more clear.

Line 538 – Triplett et al. 2014 is not formatted correctly.

Line 694 – I don’t think ROS has been defined.

Line 826 – BOP and PACMAD clades have not been defined.

Line 963 – How does the Asterids project connect with this work?

Line 968 – Specie should be species

Line 969 – “The rate was compared to that inferred from null simulation which assume no of WGD occurred.”

Line 978 – I think a word is missing : “is likely similar to the Node-Ks approach in previous studies”

Figure 1 legend: Why is the name Poaceae in green if the branches are in Red? Same with Poales? Maybe also add an explanation for $\delta^{18}O$ (‰).

Figure 7 legend: Please spell out PROSOLs.

Reviewer #1 (Remarks to the Author):

Zhang and his colleagues presented a very comprehensive profiling of the Whole-Genome Duplications (WGDs) events in Poaceae. They also reported the detailed WGD analysis in woody bamboo and *Oryza*, and finalized their work by investigating lineage specific differential retention of rho-derived duplicates and their potential functional consequences. Overall, this is a nice work that offers many novel evolutionary insights into Poaceae. One of my major concerns was that the manuscript was too long to effectively capture the key points. For example, the authors spend six pages, three sub-titles to report the major finding of the paleo-polyploidization (called kappa) event in woody bamboo, and most of the analysis in these sections were to test the topology of GDs in different lineages that supported this kappa polyploidization. I suggest the authors to re-organize these sections and simplify their writings to make this manuscript more readable.

Response: We thank the reviewer for the positive feedback. We have greatly reduced the length of descriptions in our main text, including the description about results in supplemental figures, and moved some of those descriptions to the legends of the supplemental figures to help readers to better understand our results. In particular, we have dramatically reduced the length of the part on kappa analyses.

I also have some further comments below:

1. Line 224-229, the authors applied Ks analysis to date the time of WGDs. They used the example in *Ischaemum*, with the Ks of WGD paralogs, orthologs between *Ischaemum* species and orthologs between *Ischaemum* and *Eulaliopsis* being 0.1144, 0.0599 and 0.1184, respectively. In my opinion, these data suggested that the WGD in *Ischaemum* lineage happened soon after its split with *Eulaliopsis*, so I cannot understand the author's conclusion in Line 255 that the duplications in a burst was from a single event?

Response: We are sorry for the confusion here. Our Ks comparisons agree with the reviewer's judgment in dating the WGD in *Ischaemum* lineage. More specifically, a peak in the Ks-plot of paralogs suggests a cluster of gene duplications at approximately same time in *Ischaemum* lineage. These gene duplications likely derived from a single WGD event soon after its split with *Eulaliopsis*.

To clarify the rationale of the Ks analyses, we have revised the first few sentences of this paragraph as:

“The Ks among paralogs has been widely used as a correlate of relative time for the divergence of paralogs; when Ks values form a peak in a distribution, the corresponding GDs are considered to be in a cluster near a specific time and used as support for WGDs^{1,2}. For example, the OneKP study has used detection of Ks peaks among paralogs from separate analyses of sequences of 99 single species as support for 99 WGDs in plants². Thus Ks was analyzed for paralogs identified here (see methods) and Ks peaks shared by multiple species were observed, providing additional support for WGDs from the Tree2GD analyses (Figs. S17-S23; Table S3).” (Lines 173-178)

We have also revised the sentence for our analyses in the manuscript as follows.

“In particular, the Ks peak of paralogs from a proposed WGD in a focal species is expected to have a higher value than that of orthologs between the focal species and its closely related species that also shares the WGD, and lower than that of orthologs between the focal species and an outgroup species, which diverged before the WGD event. For example, the newly proposed WGD for *Ischaemum* (WGD#13; Fig. S22; Table S3) is supported by the Ks peak value of 0.1144 for paralogs mapped at the MRCA of two *Ischaemum* species; this Ks value is higher than the Ks peak value (0.0599) of orthologs between the two *Ischaemum* species, but lower than the Ks peak value (0.1184) between *I. aristatum* and the outgroup *Eulaliopsis binata*.” (Lines 179-185)

2. The authors used the gene sequences from both genome assemblies and transcriptome assemblies to perform this phylogenomic study. As the authors have stated (Line 661), the completeness of gene annotation may affect the examination of TDs. Similarly, the completeness of transcript assemblies due to the limited RNA-seq data (Table S2) may also create bias during the identification of GDs. It's a good opportunity to evaluate the strength and weakness of using transcript assemblies to identify GDs, or at least make some discussions. By the way, I also strongly suggest the authors to divide the Results and Discussions into two different sections, and put some discussion points that currently mixed with results into the new Discussions section.

Response: We thank the reviewer for this valuable suggestion. We provided more details of our revision as below.

For discussing the GD detection using genomes and transcriptomes, we added several sentences in the manuscript (Lines 655-665) as below.

“In addition, GD detection using genomes and transcriptomes can increase the number of GDs at early branches in species-tree, because transcriptomes from the early diverging species can help to mapping gene pairs to more ancient positions. For example, among 1633 GDs mapped at Poaceae using 15 genomes, 9 transcriptomes and 2 genome-skimming datasets (Fig. S26A), 1010 and 728 GDs were shared by genes from the Anomochlooideae species *Streptochaeta angustifolia* (genome sequenced) and *S. spicata* (transcriptome sequenced), respectively. Genome-skimming sequenced datasets of the Puelioideae species (*Puelia ciliata* and *Guaduella oblongifolia*) provide genes that shared 19~36 GDs of those 1633 GDs. Genomes tend to contribute to more duplicates (423~1018 GDs) than transcriptomes (313~728 GDs) and genome-skimming datasets with incomplete sequence and annotation. Sequenced genomes also allow comparison of gene orders of paralogs on chromosomes and hence provide strong GD evidence for WGD and SSD events. Integration of genomes and transcriptomes from basal lineages to core branches can provide GD clues for understanding gene and genome evolution.”

Moreover, we divided the original section of Results and Discussions into two different sections.

3. Line 744, the authors carried out the analysis of rho-derived PROSOL genes, and reported the lineage specific PROSOL genes in Panicoideae, Pooideae, Bambusoideae, and Oryzoideae. These analyses were based on the genes that clustered into Orthogroups. Since lineage specific WGDs after rho were common in all these four subfamilies, my question was how to exclude the misidentification of rho-retained duplicates, which might actually come from the reciprocal loss of duplicate genes after lineage specific WGDs. For example, the FIE1 and FIE2 genes were reported to come from the reciprocal loss of one of their paralogs after maize WGD (Swigonova, et al., Genome Res, 2004.).

Response: We sincerely thank the reviewer for these comments to help us improve our analyses and exclude the misidentification due to lineage-specific duplication. We have performed new gene tree analyses and updated our results in main text and Figures 6 and 7. We also newly added supplemental Fig. S50-S53 and Table S5.

The two *FIE* genes in maize were reported to be two closely linked paralogous sequences (Swigonova, et al., Genome Res, 2004). To investigate the history of the *FIE1/2* genes using multiple grass genomes, we performed phylogenetic and synteny analyses of the grass *FIE* genes (including both maize *FIE1* and *FIE2*) and found that all 7 subfamilies retained the same rho-derived single copy, suggesting that their ancestor had lost the other rho-derived copy. We further found that Panicoideae (including maize and sorghum) and four other (Chloridoideae, Pooideae, Bambusoideae, and Oryzoideae) subfamilies underwent subfamily-specific duplications of *FIE* genes (Figure S50). The *FIE1/2* gene family is a good example of the lineage-specific duplicates, which should not be mis-identified as rho-derived paralogs. Our newly performed gene tree analyses allowed the identification of lineage-specific duplication in 5,666 Poaceae gene families, which have been classified as having only one rho-derived copy; the gene

family analyses also verified that 2,758 other orthogroups indeed retained both rho-derived paralogs. We have revised the context relevant to it in our new manuscript as shown below.

Lines 556-579:

“We investigated differential retention/loss patterns of rho-derived gene pairs for Poaceae subgroups. First, we detected orthogroups supported by syntenic genes from 24 sequenced grass genomes, with pineapple as an outgroup species in Poales (Fig. S48A, S48B). To obtain further support for rho-derived duplicates, we integrated our phylogenomic results (Fig. S3) and the synteny results by identifying the synteny blocks that contains at least one gene pair belonging to an orthogroup with a GD mapped at the MRCA of Poaceae or one of the early nodes with multiple subfamilies (C1-C3 in Fig. 2). To illustrate this analysis, an example of synteny blocks is shown in Fig. 6A (see details in Fig.S49A), the #6 genes correspond to a GD mapped at Poaceae in the gene tree (Fig. S49B), supporting the gene pairs (pink) in the syntenic block being from rho. The gene trees of the orthogroups were reconciled with species-tree to estimate the retention and loss events after rho in different subfamilies (see methods and supplemental Note) and the results revealed that 6,147 orthogroups retained a single copy at Poaceae and 2,758 orthogroups retained in pair (Figs. 6B and S48B, Table S5). Among the 6,147 orthogroups, 5,666 experienced subsequent lineage-specific duplication in at least one subfamily (Type-I; Fig. 6C, Table S5); one such orthogroup contains the fertilization independent endosperm (*FIE*) genes⁷³ with a duplication in Panicoideae (e.g., maize *FIE* genes [Zm00001d049608_T001(*FIE1*) and Zm00001d024698_T001(*FIE2*)]; Fig. S50). Other instances of Type-I orthogroups include the TASSELSEED2 (*TS2*)⁷⁴, DWARF53 (*D53*)⁷⁵, *COLD1*⁷⁶, and *NAC78*⁷⁷ (see details of examples in Table 1). For the remaining 481 of the 6,147 orthogroups, no more than one copy was detected in grass species (Type-II; see orthogroups in Table S5 and a specific gene tree in Fig. S51). Among 2,758 orthogroups with two rho-derived copies (Fig. 6C, Table S5), 128 (Type III) have two copies in each of the four subfamilies (Fig. S52); whereas 2,630 (Type IV) have lost one or two detected copies in at least one subfamily (see gene examples in Table 1). Among the Type IV orthogroups, we identified four patterns (IV-1 through IV-4 in Fig. 6D). Specifically, 1991 have two or more subfamilies with one detected copy (IV-1 and IV-2), including 578 that exhibit reciprocal loss of paralogs in different subfamilies (IV-2; see an example in Fig. S53). Among the 2,758 orthogroups with two copies in at least one subfamily, 565 show possible reciprocal loss of rho-derived duplicates between species within an individual subfamily (Table S5). These detected patterns should be further tested by including more high-quality genomes from different subfamilies of Poaceae and other families of Poales.”

And on lines 605-609:

“For convenience, we refer to a PROSOL specific to each of Panicoideae, Pooideae, Bambusoideae, and Oryzoideae, respectively, as PROSOL-Pa, PROSOL-Po, PROSOL-Ba, PROSOL-Or. Our examination of the above-mentioned 2,630 orthogroups in Type-IV uncovered 19 PROSOL-Pas, 8 PROSOL-Pos, 36 PROSOL-Bas, and 18 PROSOL-Ors, possibly representing subfamily-specific (or lineage-specific) retention (Fig. 7A; see representative genes in Table 2).”

Minor:

1. Line 234, “WGD dating can be affected by the ingroup species that have evolved more quickly (with greater mutation rates) than outgroup species”. Both the acceleration or decline of mutation rate may affect the estimation of molecular dating.

Response: Thanks for the suggestion. We have revised the context relevant to it in our new manuscript (Lines 191-197) as below.

“To further estimate the difference for rho, we surveyed the evolutionary rate (estimated by branch length) between species and the Ks value of retained paralogs from each species. Our results indicate that Ks values are positively correlated with the total branch length from the Poaceae MRCA to tips (Coefficient: 0.89, p-value =1.21e-08) (Fig. S24). Hence, WGD dating can be affected by the different evolutionary rates of species, including the accelerated (e.g., Panicoideae species) or reduced (e.g., Bambusoideae species) mutation rates. Thus a higher Ks peak value in a rapidly evolving lineage after

a WGD compared to the Ks peak value an outgroup that diverged before the WGD could incorrectly place a WGD at an earlier node.”.

2. Line 155, a typo of “...GDs with gene duplicates from in sequenced Poaceae genomes”

Response: We thank the reviewer for pointing the typo error out. We have revised this sentence as “). For the WGDs supported by gene duplicates from sequenced Poaceae genomes, we further estimated the number of GDs with detected duplicates in syntenic blocks.” (Lines 146-147).

3. Line 582, a typo of “...pruned the gene trees to by removing putative paralogs”

Response: Thanks for the suggestion. We have revised this sentence as “...pruned the gene trees by removing putative paralogs from duplications before the *Oryza* diversification (see methods).” (Lines 447-448).

Reviewer #2 (Remarks to the Author):

Dear Editor I have revised the manuscript titled Phylogenomic Profiles of Whole-Genome Duplications in Poaceae and Landscape of Differential Duplicate Retention and Losses among Major Poaceae Lineages by Taikui Zhang, Weichen Huang, Lin Zhang, De-Zhu Li, Ji Qi, and Hong Ma.

Whole-genome duplications (WGDs) are important drivers of angiosperms evolution and diversification. Previous studies reported three WGDs in the ecologically and economically important grass (Poaceae) family; however, these studies included a limited number of species, suspecting that there are more WGDs that has not been discovered yet. In the present work, authors performed phylogenomic/phylotranscriptomic analyses of 363 grasses covering all 12 Poaceae subfamilies to detected strong evidence for new WGDs, and other Gene Duplication clusters (GDs) and explore their possible roles in adaptive evolution and species divergence.

The paper is well written and I enjoyed the combination of methods to address, with confident, new events of WGDs and calibrate the timing of such events. From my perspective this is a very interested paper that significantly contribute with the knowledge of genome duplications presenting a whole picture of several duplication events in grasses. Sampling and methods are very well chosen and documented. Authors did an excellent selection of figures that support the main text. The manuscript presents a detailed literature compilation.

I am not a native speaker; however, in my opinion, some minor language issues need to be addressed. Specially, pay attention to large sentences.

Response: We greatly appreciate this positive comment. We have converted some long sentences to shorter ones and made other language improvements in the revised manuscript.

Reviewer #3 (Remarks to the Author):

The manuscript presents a significant contribution to the field of evolutionary biology, specifically in the realm of phylogenomics and whole-genome duplications (WGDs) in Poaceae. The findings are of considerable value to the scientific community, shedding light on lineage-specific WGDs, gene sequence evolution, and the implications of lineage-dependent duplicate retention and losses. The use of syntenic analyses to correlate gene duplicates from the same WGD event is a commendable approach. However, there are notable areas that require attention, including the need for subheaders to help clarify the paper's outline. The manuscript holds great promise and significance but requires revisions to address the outlined concerns. The incorporation of suggested improvements will contribute to the overall quality and clarity of the research.

Response: We thank the reviewer for the positive feedback. We have incorporated subheaders into the revised manuscript.

General comments:

1. Adding subheaders would enhance the manuscript's organization, providing better guidance for readers. Additionally, the outline of the paper is not entirely clear. A more structured presentation of the research would improve the overall readability. A proposed revision involves incorporating dedicated sections that accentuates some of the newly described WGD events which are discussed at length. These sections could encompass detailed methodologies employed to confirm their placements, coupled with an exploration of intriguing gene retention patterns observed during these events.

Response: We thank the reviewer for the suggestion. Because the journal does not allow the use of the secondary subheadings for results and the subheadings for discussions, we added subheaders as the first sentences for the result section and the discussion section in our new manuscript. We also listed them below.

Results

WGDs identified in Poaceae and other Poales lineages

Newly proposed WGDs and others are placed onto a species-tree and described here.

The above WGDs were also supported by evidence from Ks peaks.

Our phylogenomic analyses also detected eight other GD clusters (#23-30, Figs. S6, 7, 12, 13), in addition to the above WGD events (#1-22; Fig.1).

A proposed paleo-polyploidization of woody bamboo ancestor

We performed phylogenomic analysis with multiple species (Analysis-I) for evidence supporting Kappa. The kappa event was also supported by a phylogenomic analysis with five sequenced genomes (Analysis-II). Further support for kappa was detected from Ks analysis (Analysis-III).

Possible origin of Kappa and the Kappa-derived genes retained at tribes

Possible progenitors of Kappa were inferred.

GDs mapped to woody bamboo tribes were likely from kappa.

We further investigated the third subgenome of the hexaploid tropical woody bamboos.

Ancient hybridization contributing to the diverse adaptation of a tetraploid wild rice

Possible progenitors of the KL genomes were investigated by placing *Oryza coarctata* in the *Oryza/Oryzoideae* phylogeny.

Furthermore, we tested whether the K and L homeologs might have contributed to the adaptation of *O. coarctata*.

Differential retention and loss of rho-derived duplicates and potential functional consequences

We investigated differential retention/loss patterns of rho-derived gene pairs for Poaceae subgroups. We examined a specific pattern of differential retention/loss called the Pair Retained in One lineage but Single-copy in Other Lineages (PROSOL).

Discussions

Our combined analyses of phylotranscriptomics and chromosomal positions uncovered WGD-related gene conversion and SSDs.

Analyses here provide genomic and evolutionary insights into progenitors of WBs and a tetraploid rice. Rho-derived duplicates with different retention and loss patterns in subfamilies suggest functional evolution

2. The manuscript makes excessive use of acronyms, which may hinder comprehension for readers. It is especially difficult to keep track of the HB, WB, TWB, NWB, and PWB when discussing the ABCD genomes.

Response: We thank the reviewer for the suggestion. We have removed the acronyms of TWB, NWB, and PWB; on the other hand, we would like to retain HB and WB, which were frequently used and can save some space. We revised the relevant context in the new manuscript (Lines 299-314) as below.

“The bamboo subfamily Bambusoideae contain diploid herbaceous bamboos (HB = the Olyreae tribe; $2n=2x=20\sim 24$) and polyploid woody bamboos (WB), with the Arundinarieae (tetraploid temperate bamboos; $2n=4x=46\sim 48$) and Bambuseae tribes, which include tetraploid neotropical ($2n=4x=40\sim 48$) and hexaploid paleotropical ($2n=6x=70\sim 72$) bamboos^{11,23}. The Arundinarieae tetraploidy was supported by extensive collinearity in the sequenced *Phyllostachys edulis* genome²². In addition, a phylogenetic study⁵³ of three nuclear genes from 36 bamboo species supported a proposed 5-subgenome model (A, B, C, D and E) of WB subgenome types: AABB for Arundinarieae, CCDD for the tetraploid Bambuseae and CCDDEE for the hexaploid Bambuseae. Among the five subgenomes, B and C are relatively close. More recently, genome-scale comparison of two HBs, one Arundinarieae species, one tetraploid Bambuseae and two hexaploid Bambuseae provided support for a revised model with newly defined A, B, C, D subgenomes²³. The A subgenome is specific to hexaploid Bambuseae, B is shared by the tetraploid and hexaploid Bambuseae, C is shared by all WBs, whereas D is specific to Arundinarieae²³. Thus the hexaploid Bambuseae, the tetraploid Bambuseae, and Arundinarieae have, respectively, AABBC, BBCC, and CCDD genomes. Our above phylogenomic analyses covered five Olyreae genera, 14 genera of Arundinarieae and 14 genera Bambuseae and retrieved 6,089 GDs mapped at the MRCA of WBs, supporting a putative WGD event, namely here as Kappa (#8 in Figs. 1 and S5), providing an opportunity to examine the genome evolution pattern of WBs and to compare with the previous models. We investigated Kappa further, as described below.”

3. The manuscript should explicitly address how the current work builds on the authors previous paper on phylogeny, especially concerning the section on polyploidy. Providing this connection will enhance the contextualization of the research within the existing literature.

Response: We thank the reviewer for the suggestion. We have added several sentences in our new manuscript (Lines 139-145) as below.

“From 349 datasets (342 transcriptomes and 7 genome skimming datasets) generated for our previous grass phylogenomic/phylotranscriptomic studies^{13,20} we included 319 datasets (315 transcriptomes and 4 genome skimming datasets) to detect WGDs. The analyses here included the use of gene- and species-tree reconciliation using Tree2GD and Ks analyses (see species-tree in Figs. S1 and S2, and more details in methods), taking advantage of the recently established Poaceae/Poales phylogenies¹³. Additional datasets for 51 Poaceae, 10 for other Poales, and 10 for other orders were retrieved from public databases (see taxon and transcript assembly information in Table S1 and Table S2, respectively).”

4. I really appreciate the authors detailed explanation of the PROSOL genes, but I was wondering if this could possibly be condensed to a table with a shorter summary in the main text. Currently, the paper is over 30 pages, and this could be a way to shorten it.

Response: We thank the reviewer for the suggestion. We have shortened the text length by moving most of the gene information to Tables 1 and 2.

Specific comments:

Line 15 – the first sentence of the abstract is hard to follow, please consider breaking into multiple sentences.

Response: Thanks. We have revised this sentence in new manuscript as shown below.

“Whole-genome duplications (WGDs) are important for angiosperm genome and gene evolution. Poaceae are the fifth largest plant family with ~12,000 species, including major crops (rice, wheat, and maize) and species with diverse habits and morphologies.” (Lines 15-17).

Line 43 – I don’t think the authors mean molecular dating placed the WGD events, but rather some other analyses did?

Response: Thanks. We have revised this sentence in new manuscript (Lines 42-44) as below.

“In particular, analyses of synonymous substitution rate (K_s) values of gene duplicates placed 61 angiosperm WGDs on branches with increased diversification rates, suggesting importance of WGDs in diversification⁸.”

Line 88 – “Additionally, rice and other *Oryza* species collectively have 11 reported genome types (six diploids and five allotetraploids); furthermore, domestic and wild *Oryza* species that have adapted to different aquatic environments.” – this sentence could be broken up for clarity.

Response: Thanks. We have revised this sentence in new manuscript (Lines 82-84) as shown below.

“Additionally, rice and other *Oryza* species collectively have 11 reported genome types (six diploids and five allotetraploids)²⁷. Domestic and wild *Oryza* species have adapted to different aquatic environments¹⁴.”

Line 97 – Can the authors please clarify what they mean “successive species phylogenetic positions”

Response: Thanks. We have revised this sentence in new manuscript (Line 90) as shown below.

“This idea is supported by the detection of GD clusters at successive nodes on species trees in phylogenomic studies^{1,9,36}.”

Line 129 – this paragraph ends with recently and the next paragraph starts with the same word.

Response: We have removed the second “recently” at the beginning of the next paragraph.

Line 155 – Can the authors please clarify what they mean by “For those WGDs that are supported by GDs with gene duplicates from in sequenced Poaceae genomes, we further estimated the number of GDs with detected duplicates in syntenic blocks (collinear genomic regions).”

Response: We are sorry for the confusion. Also in response to a comment from reviewer#1, we have revised this sentence as below.

We have revised this sentence as “For the WGDs supported by gene duplicates from sequenced Poaceae genomes, we further estimated the number of GDs with detected duplicates in syntenic blocks.” (Lines 146-147).

Line 216 – I am not sure that Ks analyses can be referred to as molecular dating. I may be incorrect, but they are all relative, not absolute, so perhaps avoiding calling them molecular dating would be more clear.

Response: Thanks. We have revised this sentence in new manuscript (Line 173-175) as follows.

“The Ks among paralogs has been widely used as a correlate of relative time for the divergence of paralogs; when Ks values form a peak in a distribution, the corresponding GDs are considered to be in a cluster near a specific time and used as support for WGDs^{1,2}.”

Line 538 – Triplett et al. 2014 is not formatted correctly.

Response: Thanks. Because the description of WGD analyses of bamboos was reduced, this portion of the sentence was removed.

Line 694 – I don't think ROS has been defined.

Response: Thanks. We have revised it in the new manuscript (Line 539-540) as shown below.

“Peroxidase genes belong to a superfamily in plants and play key roles in reactive oxygen species (ROS) formation⁷¹.”

Line 826 – BOP and PACMAD clades have not been defined.

Response: Thanks. We revised them in our new version of manuscript (Lines 52-56) as follows.

“Poaceae are the fifth-largest family (~12,000 species in 12 subfamilies) and include numerous economically important species in Panicoideae (maize and sorghum; the 2nd largest) and Chloridoideae (teff) of the PACMAD clade (also including Danthonioideae, Arundinoideae, Micrairoideae, Aristidoideae), and in subfamilies of the BOP clade consisting of Pooideae (wheat and barley; the largest subfamily), Oryzoideae (rice), and Bambusoideae (bamboos, the 3rd largest) (e.g., ref¹¹⁻¹³).”

Line 963 – How does the Asterids project connect with this work?

Response: Thanks. We are sorry that the original sentence was not clear and have revised the sentence in the new version of manuscript (Lines 803-806) as shown below.

“To measure the statistically significant difference between the ratio of retained GDs from ancient WGDs and SSDs, we analyzed the rho, sigma and tau events, according to the minimum number of required species in the ladder species-tree for MAPS and also other effective estimation for ancient WGDs in previous analyses by using MAPS^{1,2,48} (see detailed procedures in Supplementary Note).”

Line 968 – Specie should be species

Response: Thanks. Done.

Line 969 – “The rate was compared to that inferred from null simulation which assume no of WGD occurred.”

Response: Thanks. We have changed the sentence to: “The rate was compared to that inferred from null simulation which assumed no occurrence of WGD.”. (Lines 810-811)

Line 978 – I think a word is missing : “is likely similar to the Node-Ks approach in previous studies”

Response: Thanks. We have revised this sentence in the revised manuscript (Lines 817-819) as shown below.

“To infer candidate WGDs in each taxon with transcriptome or low-coverage genome datasets, we utilized a method that has been included in Tree2GD⁴ and is similar to the Node-Ks approach used in previous studies^{1,2} (see details in Supplementary Note).”

Figure 1 legend: Why is the name Poaceae in green if the branches are in Red? Same with Poales? Maybe also add an explanation for $\delta 18\text{O}$ (‰).

Response: We are sorry for the color differences of the taxon names and branches; we have changed the color of the name Poaceae to red and that of the name Poales to green. We also thank the reviewer for the suggestion of $\delta 18\text{O}$ and explained it in the new legend as follows.

“The red curve in the graph below the stratigraphic boxes indicates the changes in oxygen isotope records of $\delta 18\text{O}$ (‰) (the left Y-axis), reflecting the temperature changes as indicated by the right Y-axis.” (Lines 1188-1190)

Additionally, to help readers better understand the $\delta 18\text{O}$ information, we added sentences to the ‘Bayesian divergence time estimation’ section in Supplementary Note as shown below.

“The time-tree was compared with the paleo-climate changes during stratigraphic periods. The free-ice temperature was estimated by using the oxygen isotope $\delta 18\text{O}$ content in fossils, which has been applied as a method to reflect the climate changes on geologic time-scales³⁵. The $\delta 18\text{O}$ data were retrieved from previous analyses^{35,36}.”

Figure 7 legend: Please spell out PROSOLs.

Response: Thanks. We have revised it as below.

“Figure 7 Pair Retained in One lineage but Single-copy in Other Lineages (PROSOLs) and functional implications” (Line 1309)

Reviewers' Comments:

Reviewer #1:

Remarks to the Author:

The authors have addressed my previous concerns, and the manuscript has been significantly improved. I believe this work will be an important advance in the field of crop evolutionary genomics. I recommend its publication in Nat Commun.

Reviewer #3:

Remarks to the Author:

The revised manuscript of Zhang et al. has been greatly improved. The addition of subheaders and topic sentences greatly improves the readability of the manuscript. However, there are still some remaining acronyms which are not defined (see examples below) and some clarification needed in some places.

Below are some minor comments that I believe can be addressed easily and should not hold up acceptance of the manuscript.

Line 30 – there is an extra "and" at the end of the line I believe.

Line 56 – Perhaps the sentence, "The two clades comprise the core Poaceae" could be edited to refer back to the PACMAD and BOP clades? Or some sort of reorganization that notes that the Poaceae are comprised of two clades named PACMAD and BOP. As this is currently written it is hard to understand to a non-Poaceae expert.

Line 140 – Are the 319 datasets newly generated for this study? It is still unclear.

Line 288 – *B. rapa* and *B. oleracea* are not needed at the end of this sentence as they are referred to at the start of the sentence.

Line 378 – LBA is not defined.

Line 425 – *Oryza* should be spelled out at the start of a sentence.

Line 657 – mapping should be changed to map.

Line 674 – Higher levels is incorrect, please correct to "some seed plants".

Line 792 – Should this say "place GDs on the species-tree"?

Line 868 – Please redefine HB and WB in the methods section.

Line 884 – what does SWS and PWS mean?

In general, there is an inconsistent use of oxford commas.

Reviewer #1 (Remarks to the Author):

The authors have addressed my previous concerns, and the manuscript has been significantly improved. I believe this work will be an important advance in the field of crop evolutionary genomics. I recommend its publication in Nat Commun.

Response: We greatly appreciate this positive comment.

Reviewer #3 (Remarks to the Author):

The revised manuscript of Zhang et al. has been greatly improved. The addition of subheaders and topic sentences greatly improves the readability of the manuscript. However, there are still some remaining acronyms which are not defined (see examples below) and some clarification needed in some places.

Response: We greatly thank the reviewer for the positive feedback. We have revised the acronyms and clarification in our new manuscript. Please find our response to relevant revisions below.

Below are some minor comments that I believe can be addressed easily and should not hold up acceptance of the manuscript.

Line 30 – there is an extra "and" at the end of the line I believe.

Response: Thanks. We have revised this sentence in new manuscript as shown below.

“Moreover, rho duplicates showing differential retention among subfamilies include those with functions in environmental adaptations or morphogenesis, including *ACOT* for aquatic environments (Oryzoideae), *CK2β* for cold responses (Pooideae), *SPIRALI* for rapid cell elongation (Bambusoideae), and *PAIL* for drought/cold responses (Panicoideae).” (Lines 22-25)

Line 56 – Perhaps the sentence, “The two clades comprise the core Poaceae” could be edited to refer back to the PACMAD and BOP clades? Or some sort of reorganization that notes that the Poaceae are comprised of two clades named PACMAD and BOP. As this is currently written it is hard to understand to a non-Poaceae expert.

Response: We thank the reviewer for the suggestion. We revised the relevant context in the new manuscript (Lines 44-49) as shown below.

“Poaceae are the fifth-largest family (~12,000 species in 12 subfamilies) and the core Poaceae comprise two clades named PACMAD and BOP¹¹⁻¹³. The PACMAD clade includes subfamilies Panicoideae, Chloridoideae, Danthonioideae, Arundinoideae, Micrairoideae, and Aristidoideae. The BOP clade consists of subfamilies Pooideae, Oryzoideae, and Bambusoideae. Poaceae include numerous economically important species in Panicoideae (maize and sorghum; the 2nd largest subfamily), Chloridoideae (teff), Pooideae (wheat and barley; the largest), Oryzoideae (rice), and Bambusoideae (bamboos, the 3rd largest) (e.g., ref¹¹⁻¹³).”

Line 140 – Are the 319 datasets newly generated for this study? It is still unclear.

Response: We thank the reviewer for the suggestion. We revised the relevant context in the new manuscript (Lines 135-137) as shown below.

“Among the published 349 datasets (342 transcriptomes and 7 genome skimming datasets) generated for our previous grass phylogenomic/phylotranscriptomic studies^{13,20}, we selected 319 datasets (315 transcriptomes and 4 genome skimming datasets) for our analyses here (Supplementary Data 1).”

Line 288 – B. rapa and B. oleracea are not needed at the end of this sentence as they are referred to at the start of the sentence.

Response: Thanks. We have revised this sentence in our new manuscript (Lines 294-295) as shown below.

“In addition, comparison of *Brassica rapa* and *B. oleracea* sequences supported gene conversion of 368 and 343 syntenic genes, respectively⁵¹.”

Line 378 – LBA is not defined.

Response: Thanks. We have revised it in our new manuscript (Lines 395-396) as shown below.

“These results suggest that long branch attraction could not fully explain the detected GDs at the MRCA of Bambusoideae.”

Line 425 – *Oryza* should be spelled out at the start of a sentence.

Response: Thanks. We have revised it as ‘*Oryza*’.

Line 657 – mapping should be changed to map.

Response: Thanks. Done.

Line 674 – Higher levels is incorrect, please correct to “some seed plants”.

Response: Thanks. We have revised it in our new manuscript (Lines 701-704) as below.

“Our phylogenomic/phylotranscriptomic analyses detected GD clusters at successive nodes on the species phylogeny, as also observed in other WGD analyses of different taxonomic groups, including genera, subtribes, tribes, subfamilies, families, orders, and at broader scales, such as angiosperms and other seed plants (e.g., ref^{1,2,36,48,91}).”

Line 792 – Should this say “place GDs on the species-tree”?

Response: Thanks. We have revised it in our new manuscript (Lines 882-883) as shown below.

“To identify WGD events shared by two or more species and place the published WGDs to the species tree, we used Tree2GD v1.0.40⁴ to place GDs on the species-tree.”

Line 868 – Please redefine HB and WB in the methods section.

Response: Thanks. We have revised the relevant context in our new manuscript (Lines 1064-1066) as shown below.

“To detect possible genome regions related to hybridization, using above MCScan approach, the *O. latifolia* genome was aligned against the *P. edulis* or *D. latiflorus* genomes to identify inter-species collinear blocks between herbaceous bamboos (HB) and woody bamboos (WB) (Supplementary Fig. 35).”

Line 884 – what does SWS and PWS mean?

Response: Thanks. We have revised them in our manuscript (Lines 1087-1089) as shown below.

“The RNA-Seq data⁶⁴ of *O. coarctata* under control (SRR771527), under salt-water submergence (SRR771531), and under purified-water submergence (SRR771530) (Supplementary Data 1) are used for calculating transcripts per million values by kallisto v0.46.1¹⁶¹.”

In general, there is an inconsistent use of oxford commas.

Response: We greatly thank you for the comment. We have prof-read our manuscript to revise the oxford commas. Below are revised lists.

“The PACMAD clade includes subfamilies Panicoideae, Chloridoideae, Danthonioideae, Arundinoideae, Micrairoideae, and Aristidoideae.”

“...six other WGDs (#9, 11, 12, 14, 17, and 18; Supplementary Figs. 8, 9, 11, 13, 16, and 14).”

“...detected in Type-I (such as basal transcription machinery binding, lipid transporter activity, and signaling receptor binding)”

“For convenience, we refer to a PROSOL specific to each of Panicoideae, Pooideae, Bambusoideae, and Oryzoideae, respectively, as PROSOL-Pa, PROSOL-Po, PROSOL-Ba, and PROSOL-Or.”

“...gene sets to contain *Amborella trichopoda* for crown angiosperms, *Zostera marina* for crown monocots, *Ananas comosus* for crown Poales, and *Streptochaeta* for crown Poaceae.”